# Dietary change without caloric restriction maintains a youthful profile in ageing yeast

**Dorottya Horkai**[¤], **Hanane Hadj-Moussa, Alex J. Whale, Jonathan Houseley***

Epigenetics Programme, Babraham Institute, Cambridge, United Kingdom

¤ Current address: Abcam plc, Biomedical Campus, Cambridge, United Kingdom
* jon.houseley@babraham.ac.uk

**Data Availability Statement:** All RNA-seq data has been deposited at GEO under accession number GSE207503. All other relevant data are within the paper and its Supporting information files. Detailed and updated protocols are available at https://www.

## Abstract

Caloric restriction increases lifespan and improves ageing health, but it is unknown whether these outcomes can be separated or achieved through less severe interventions. Here, we show that an unrestricted galactose diet in early life minimises change during replicative ageing in budding yeast, irrespective of diet later in life. Average mother cell division rate is comparable between glucose and galactose diets, and lifespan is shorter on galactose, but markers of senescence and the progressive dysregulation of gene expression observed on glucose are minimal on galactose, showing that these are not intrinsic aspects of replicative ageing but rather associated processes. Respiration on galactose is critical for minimising hallmarks of ageing, and forced respiration during ageing on glucose by overexpression of the mitochondrial biogenesis factor Hap4 also has the same effect though only in a fraction of cells. This fraction maintains Hap4 activity to advanced age with low senescence and a youthful gene expression profile, whereas other cells in the same population lose Hap4 activity, undergo dramatic dysregulation of gene expression and accumulate fragments of chromosome XII (ChrXIIr), which are tightly associated with senescence. Our findings support the existence of two separable ageing trajectories in yeast. We propose that a complete shift to the healthy ageing mode can be achieved in wild-type cells through dietary change in early life without caloric restriction.

## Introduction

Dietary restriction can improve ageing health in eukaryotes ranging from budding yeast to primates and even humans, but remains an unrealistic approach for improving ageing health in human populations as this sacrifice requires substantial commitment [1–5]. However, the beneficial effects of dietary restriction are not simply due to a reduction in calories as equivalent effects can be achieved through limiting or even tailoring amino acid intake, indicating that improved ageing health results from altered metabolic state rather than minimal energy intake [6–8]. Improved ageing health may therefore be achievable in humans through optimising diet, but this will require a much better understanding of the interplay between diet and the biological mechanisms leading to ageing pathology [9].

babraham.ac.uk/our-research/epigenetics/jon-houseley/protocols.

**Funding:** JH was funded by the Wellcome Trust [110216], DH by a DTP PhD award [1947502], JH, HHM, AW and DH by the BBSRC [BI Epigenetics ISP: BBS/E/B/000C0423]. The funders had no role in study design, data collection and analysis, decision to publish, or preparation of the manuscript.

**Competing interests:** The authors have declared that no competing interests exist.

**Abbreviations:** ERC, extrachromosomal rDNA circle; MEP, mother enrichment program; ncRNA, non-coding RNA; PCA, principal component analysis; rDNA, ribosomal DNA; ROS, reactive oxygen species; SEP, senescence entry point; WGA, wheat germ agglutinin.

Replicative ageing of the budding yeast *Saccharomyces cerevisiae* is routinely employed as a rapid and genetically tractable model system for ageing research. Budding yeast cells divide asymmetrically into a mother cell and a daughter cell, with the mother cell undergoing only a limited number of divisions before permanently exiting the cell cycle [10]. Mother cells maintain a rapid and uniform cell cycle for most of their replicative lifespan but the cell cycle slows dramatically in the last few divisions [10–12], a point designated the senescence entry point (SEP) that is coincident with the appearance of apparently pathological markers including mitochondrial protein foci formation and nucleolar expansion [11,13]. Many other changes occur during replicative ageing, including cell growth [10,14], loss of mitochondrial membrane potential [15], mitochondrial fragmentation [16], loss of vacuolar acidity [17], nuclear pore disruption [18], and protein oxidation and aggregation [19,20], each of which likely contributes to the eventual loss of replicative potential, but how these relate to each other or to the extension of cell cycle timing at the SEP remains largely unknown.

Genetic and epigenetic instability are hallmarks of ageing, and both processes are observed during yeast replicative ageing [21]. Although the yeast genome does not accrue significant mutations with age, recombination events within the ribosomal DNA (rDNA) give rise to extrachromosomal rDNA circles (ERCs), which accumulate massively with age to add 30% to 40% to total genome size in a single day [22–24]. Chromatin structure and the genomic distribution of histone modifications also change dramatically with age, either as cause or consequence of gene expression change [24–26]. Progressive dysregulation of gene expression accompanies yeast replicative ageing [26–30], with consequent remodelling of the proteome and metabolome in a manner consistent with activation of the environmental stress response [28,30–33]. However, it is not known whether these changes are pathogenic or protective, nor is it clear whether gene expression dysregulation is a consequence of genomic change.

Recent studies have revealed heterogeneity in ageing trajectory, with cells following either a slow dividing, short-lived trajectory marked by up-regulation of iron transporters and accumulation of the Tom70-GFP mitochondrial protein foci linked to SEP onset or a fast dividing, long-lived trajectory, marked by nucleolar expansion and weak rDNA silencing [34–36]. The association of weak rDNA silencing with a long-lived trajectory is surprising as rDNA silencing mutants such as *sir2Δ* undergo higher rates of rDNA recombination and ERC formation due to interference between non-coding RNA (ncRNA) transcription and sister chromatid cohesion [37–40]. Furthermore, other studies find nucleolar expansion to be linked to onset of the SEP, defects in ribosome precursor export and low iron transporter induction [13,41]. Therefore, the link between mitochondrial dysfunction and senescence is supported across studies, but nucleolar changes are more variable across experimental systems and the importance and timing of rDNA instability remains to be clarified. A major contributor to these differences may be variation in rDNA copy number between strains, which changes during routine yeast transformation and influences both lifespan and rDNA recombination rate [42,43].

Dietary restriction is most commonly achieved in yeast through caloric restriction—a reduction in media glucose concentration from 2% to 0.5% that reproducibly increases lifespan [44]. This has been ascribed both to a reduction in growth signalling through PKA and TOR pathways [44,45] and an induction of respiration acting through Sir2 to improve rDNA stability and decrease ERC formation [46–49]. Unfortunately, it is technically very difficult to purify yeast aged under caloric restriction so the long-term physiological effects of caloric restriction remain largely unexplored. Nonetheless, cells aged under caloric restriction do show increased cell cycle time at the end of replicative lifespan, suggesting that the SEP is not repressed [50]. Unexpectedly, we have noted that changing the primary carbon source during yeast ageing improves the homogeneity of cycling times in aged cells [51]. We suggest that maintenance of a rapid uniform cell cycle provides the best definition of ageing health in

budding yeast, and thereby, ageing health must decline dramatically at the SEP, which can be assayed using known SEP-associated markers. Here, we demonstrate that growth on galactose as an alternate carbon source in early life dramatically improves ageing health of yeast through enhanced respiration without increasing lifespan and suppresses both gene expression dysregulation and genome instability.

## Results

### Suppression of age-linked phenotypic change on an unrestricted galactose diet

*S. cerevisiae* preferentially metabolise glucose, but the fermentable sugars galactose and fructose also support robust growth in culture. To determine the effect of non-preferred carbon sources on ageing, we used an optimised mother enrichment program (MEP) protocol in which cells are biotin labelled then aged in rich media for 24 or 48 h, fixed and mother cells purified (Fig 1A) [24,52], and ensured by bud scar counting that cells aged in glucose and galactose reached equivalent average replicative ages at 24 and 48 h (S1A Fig). The MEP system renders daughter cells inviable, such that mother cells can reach very advanced age without media components being exhausted by proliferation of offspring.

The SEP represents an abrupt transition in ageing at which yeast cells cease to divide rapidly and the cell cycle becomes slow and heterogeneous [11]. Post-SEP cells are marked by highly fluorescent foci of the GFP-tagged mitochondrial protein Tom70 in G1 [11,35]. To quantify Tom70-GFP fluorescence as an SEP marker, we optimised an imaging flow cytometry assay for aged MEP cells expressing Tom70-GFP as well as an Rpl13a-mCherry marker that accumulates only slightly with age [53]. Per sample, 1,000 purified mother cells gated for streptavidin and roundness are imaged for brightfield, Tom70-GFP, Rpl13a-mCherry, and WGA-AF405 to label bud scars (Fig 1A, right).

Tom70-GFP increased substantially with age across the whole population in glucose (Fig 1B, left, glucose), which was expected as the roundness gate selects towards G1 cells in which Tom70-GFP foci manifest [11]. In contrast, Rpl13a-mCherry increased slightly but became much more heterogeneous at 24 h and often declined from 24 to 48 h, perhaps reflecting the onset of ribophagy once cells cease to divide (Fig 1B, middle, glucose). Wheat germ agglutinin (WGA) fluorescence increased progressively and correlated to the accumulation of bud scars on both glucose and galactose (S1A Fig), forming a useful metric for replicative age as previously noted [28,54,55]. Imaging flow cytometry therefore allows rapid quantification of age and age-associated phenotypes, with excellent reproducibility across biological replicates (Fig 1C).

Remarkably, when glucose was replaced with galactose, the Tom70-GFP and Rpl13a-mCherry markers behaved very differently: Tom70-GFP increased only marginally with age (Fig 1B and 1C, left), while the Rpl13a-mCherry signal increased at 24 h and remained high from 24 to 48 h without becoming heterogeneous (Fig 1B and 1C, middle). These differences suggest that ageing of MEP diploid cells in galactose is not associated with an SEP, despite cells reaching the same replicative age after 24 and 48 h as measured by both WGA fluorescence intensity and manual bud scar counting (Fig 1C, right and S1A Fig). Mitochondrial dysfunction in aged cells is associated with defects in the vacuole [17,56], so we examined the vacuolar marker Vph1-mCherry to reinforce the mitochondrial Tom70-GFP phenotype. Vph1-mCherry intensity and area increased substantially with age in glucose, showing that the Tom70-GFP phenotype is not an artefact of a single protein or fluorescent tag, but Vph1-mCherry intensity was almost unchanged with age in galactose (S1B Fig, left), while the increase in vacuole size was significantly lessened (S1B Fig, right). Therefore, ageing

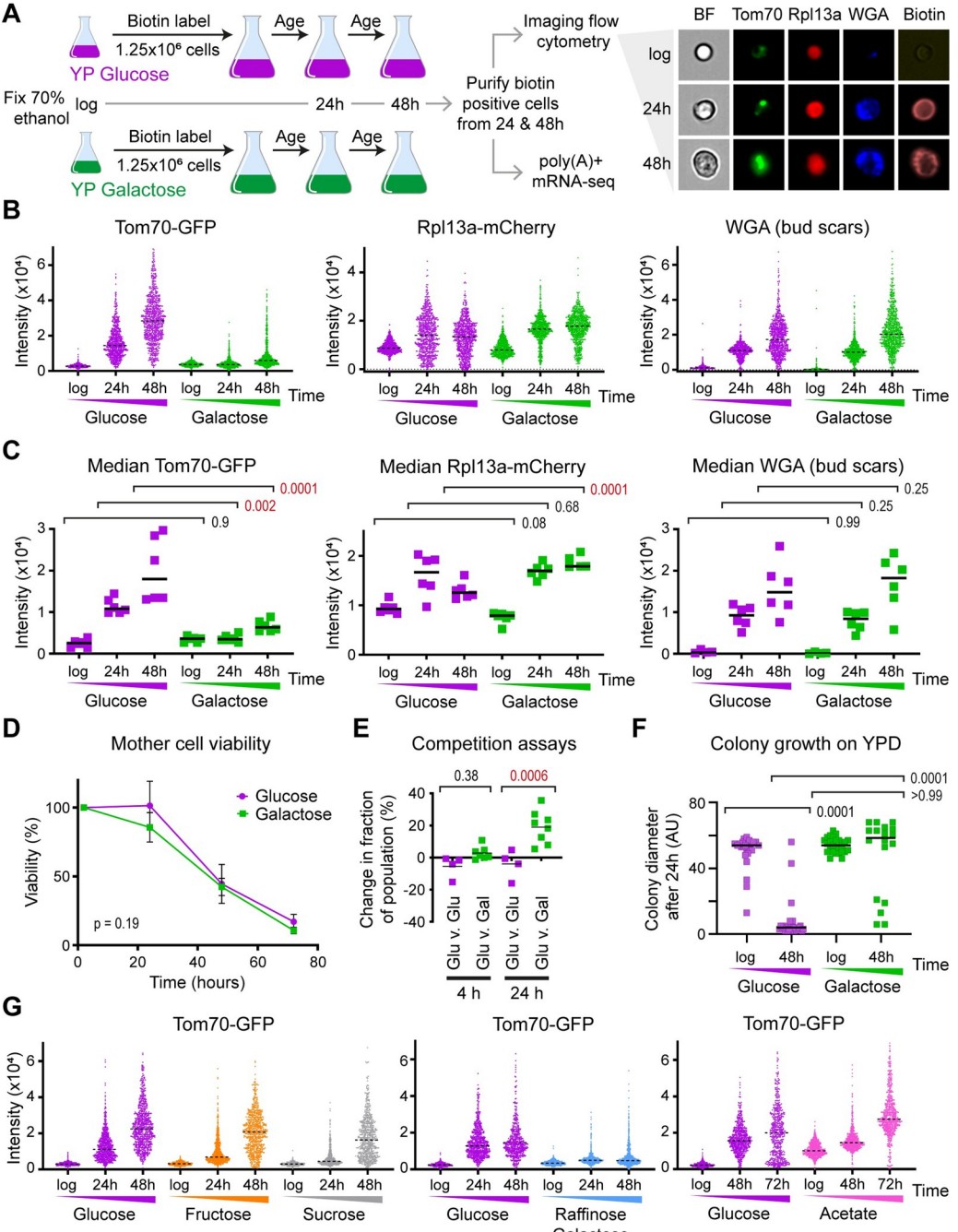

**Fig 1. Progression of age-linked phenotypes on different diets.** (A) Schematic of ageing cell culture and analysis, with examples of imaging flow cytometry data (Tom70-GFP, Rpl13a-mCherry, WGA-Alexa405, and brightfield) for wild-type MEP cells ageing on YPD (rich glucose media). (B) Distribution of signal intensities from flow cytometry images of Tom70-GFP, Rpl13a-mCherry, and WGA-Alexa405 in populations of log phase cells or mother cells aged for 24 and 48 h in YP with 2% glucose or galactose. A total of 1,000 cells are imaged per population after gating for circularity (all) and biotin (aged cells only, based on streptavidin-Alexa647). Images are post-filtered for focus of Tom70-GFP, leaving approximately 600 cells per population. (C) Median signal intensities from multiple biological replicates of populations described in B. $n = 6$, $p$ values calculated by two-way ANOVA with post hoc Tukey test for age and diet. (D) Lifespan measured in YPD and YPGal media based on %age of viable cells remaining at each time point; p value from t test comparing area under curves, $n = 5$ for glucose and for galactose. (E) Competition assays measuring fitness in glucose between cells aged for indicated times on glucose or galactose. Wild-type MEP cells carrying G418 or Nourseothricin resistance genes were aged for 4 and for 24 h post estradiol addition in YP media with 2% glucose or galactose, then mixed in pairs carrying reciprocal markers for competition and plated on G418 and on nourseothricin plates at t = 0 to

determine initial mixture competition, then inoculated at 1:1,000 in YPD and allowed to grow to saturation before plating again on G418 and on nourseothricin plates. Change in composition (%) from t = 0 to end of competition is plotted on y-axis. $n = 4$ glucose v glucose, $n = 8$ glucose v galactose, $p$ values calculated by one-way ANOVA with post hoc Tukey test. (F) Colony size assays measuring fitness of cells aged in glucose or galactose. Cells were aged 48 h in glucose or galactose, purified live in media then placed on YPD plates by micromanipulation. Colony diameters were measured on the micromanipulator screen after 24 h. Only cells that formed visible colonies after a further 3 days were included in the analysis. Log phase cells were taken directly from culture and colony growth measured under the same conditions; $p$ values calculated by Kruskal–Wallis test; $n = 25$ for log glucose, log galactose, and aged glucose; $n = 19$ for aged galactose. (G) Population distributions of Tom70-GFP for cells aged on indicated carbon sources, matching Rpl13a-mCherry and WGA data shown in S1D and S1E Fig. Note extended time-course glucose control for acetate experiment, however, 72 h acetate is age-matched to 48 h glucose as cell division is slower on acetate. The data underlying this figure can be found in S2 File. MEP, mother enrichment program; WGA, wheat germ agglutinin.

phenotypes revealed by multiple markers are suppressed when cells are aged on galactose instead of glucose.

The glucose-galactose differences could reflect a dramatic extension of lifespan such that the SEP is simply delayed, but the viability of cells on glucose and galactose decayed at a similar rate over time in culture (Fig 1D). Given that bud scar counts and WGA intensity at 24 and 48 h are equivalent, lifespan cannot be substantially different between glucose and galactose (Figs 1C and S1A), in accord with a previous report that lifespan is 17% shorter on galactose [57]. Alternatively, these markers may not accurately report cellular fitness; however, when directly competed in glucose media, cells aged for 24 h in galactose out-perform cells equivalently aged in glucose (Fig 1E). Furthermore, although competition assays are difficult at 48 h due to low cell viability, most cells aged 48 h in galactose formed colonies on YPD plates of equivalent size to those formed by young cells, whereas all colonies formed by cells aged 48 h in glucose were smaller, and most of these cells divided 2 or less times in the first 24 h after plating, showing that cells aged on glucose divide slowly whereas cells aged on galactose do not (Fig 1F). This means that despite having a similar or shorter lifespan, diploid MEP cells aged on galactose do not show an SEP and remain fitter than those aged on glucose.

In contrast, fructose did not strongly suppress ageing phenotypes: Tom70-GFP was somewhat reduced at 24 h and 48 h relative to glucose ageing but was still higher than at log phase and the difference was only significant at 24 h (S1C Fig). Furthermore, the behaviour of the Rpl13a-mCherry marker was indistinguishable between glucose and fructose (S1C Fig). This indicated that different diets have different effects on ageing in yeast so we surveyed other carbon sources (Figs 1G and S1D): sucrose acted very similarly to fructose, a 2% raffinose 0.5% galactose mixture was similar to galactose alone, while acetate was exceptional as Tom70-GFP accumulated similarly to cells in glucose but Rpl13a-mCherry did not decrease at advanced ages. Cells aged in fructose, sucrose, and raffinose/galactose attained equivalent replicative age at 24 and 48 h to cells aged in glucose (S1E Fig), but cell division was slow in acetate, which prevents like-for-like comparison (S1E Fig), and mitochondrial biogenesis is high which will contribute to the Tom70-GFP signal independent of the SEP.

Taken together, these experiments reveal that cells aged in galactose undergo minimal age-linked changes based on fluorescence markers of the SEP and direct measurement of fitness. This occurs without an increase in lifespan, proving that at least in yeast, lifespan is distinct from senescence and loss of fitness.

## The transcriptome undergoes minimal dysregulation during ageing in galactose

Many studies have reported characteristic changes in gene expression patterns during yeast replicative ageing on glucose [24,26–30]. This may represent a response to metabolic stresses

arising during ageing (for example, activation of the environmental stress response), dysregulation of transcription, or both. The dramatic differences in ageing phenotype between glucose and other carbon sources led us to examine the behaviour of the transcriptome, both as an orthogonal marker of the ageing process and to provide mechanistic insights.

Distortion of global gene expression is easily visualised on an MA plot, which for each mRNA plots the change in abundance between 2 conditions against average abundance. Comparing young and old cells in glucose shows that genes with low average expression undergo large increases in abundance with age, whereas highly expressed genes (mostly ribosome and ribosome biogenesis proteins) are slightly reduced relative to average, resulting in the distribution becoming skewed away from the x-axis in accord with previous studies (Fig 2A, left) [24,26–29]. However, comparison of young and old cells in galactose reveals gene expression changes but only a weak bias towards induction of normally inactive genes between log and 48 h (Fig 2A, right), showing that age-linked dysregulation of mRNA abundance is not an intrinsic aspect of the ageing process. Equivalent results were obtained comparing mothers harvested after 7.5 and 48 h (S2A Fig), showing that the observed differences between glucose and galactose are not driven by the large fraction of newborn daughter cells in the log phase samples.

Transcriptomes of cells aged on fructose, sucrose, raffinose/galactose, and acetate show varying degrees of dysregulation (S2B Fig) that we captured in a single value—the slope of a line of best fit drawn from an MA plot comparing young and old. These lines (Figs 2A and S2) have a slope approaching zero in galactose where there is little directional bias in the gene expression changes during ageing, but become increasingly negative with age-linked gene expression dysregulation. This slope value correlates very well to Tom70-GFP intensity across different ages and media conditions (Fig 2B), indicating that transcriptional dysregulation is associated with the onset of the SEP. The only exception is acetate, which has a low slope but a high Tom70-GFP signal that likely reflects very high mitochondrial biogenesis on this strict oxidative phosphorylation substrate rather than SEP-associated Tom70 accumulation.

Principal component analysis (PCA) of this transcriptome data from cells aged on different carbon sources revealed that samples separated by replicative age on the first principal component (PC1, explaining 45% of the dataset variance) and largely by carbon source on PC2 (explaining 21% of variance) (Fig 2C). Glucose and fructose, which are metabolised similarly, cluster together at log and 48 h time points, with fructose 24 h lagging slightly behind glucose 24 h on the same trajectory (Fig 2C, purple and orange). The transcriptional profile associated with sucrose fermentation differs from glucose and fructose and as such the log phase sucrose profile is separated on PC2 [58]; however, sucrose is processed into glucose and fructose prior to being metabolised, and aged sucrose transcriptomes are very similar to fructose (Fig 2C, grey). Galactose-grown samples migrate less distance on PC1 with age, are separated from glucose and fructose on PC2 at all time points, and change little between 24 h and 48 h showing that gene expression is relatively stable (Fig 2C, green), with the raffinose/galactose mixture following the same trajectory (Fig 2C, blue). Acetate is again the exception as it also moves little on PC1 with age but even at log phase is already coincident on PC1 with highly aged samples from other carbon sources, probably reflecting very slow growth rate (Fig 2C, pink). PC1 loading is dominated by increasing cell wall synthesis and decreasing ribosome synthesis, while PC2 loading is driven by increasing respiration and decreasing retrotransposition.

Ribosome synthesis is tightly coupled to growth rate [59], suggesting that growth rate is a major contributor to PC1; although we observe that mother cell cycle rate is equivalent between glucose and galactose, population doubling time in galactose is longer indicating slower growth of newborn daughters. Indeed, log phase samples on glucose are located more towards the high ribosome synthesis/translation end of PC1 than log phase cells on galactose, reflecting faster population growth. Nonetheless, highly aged cells on glucose are located

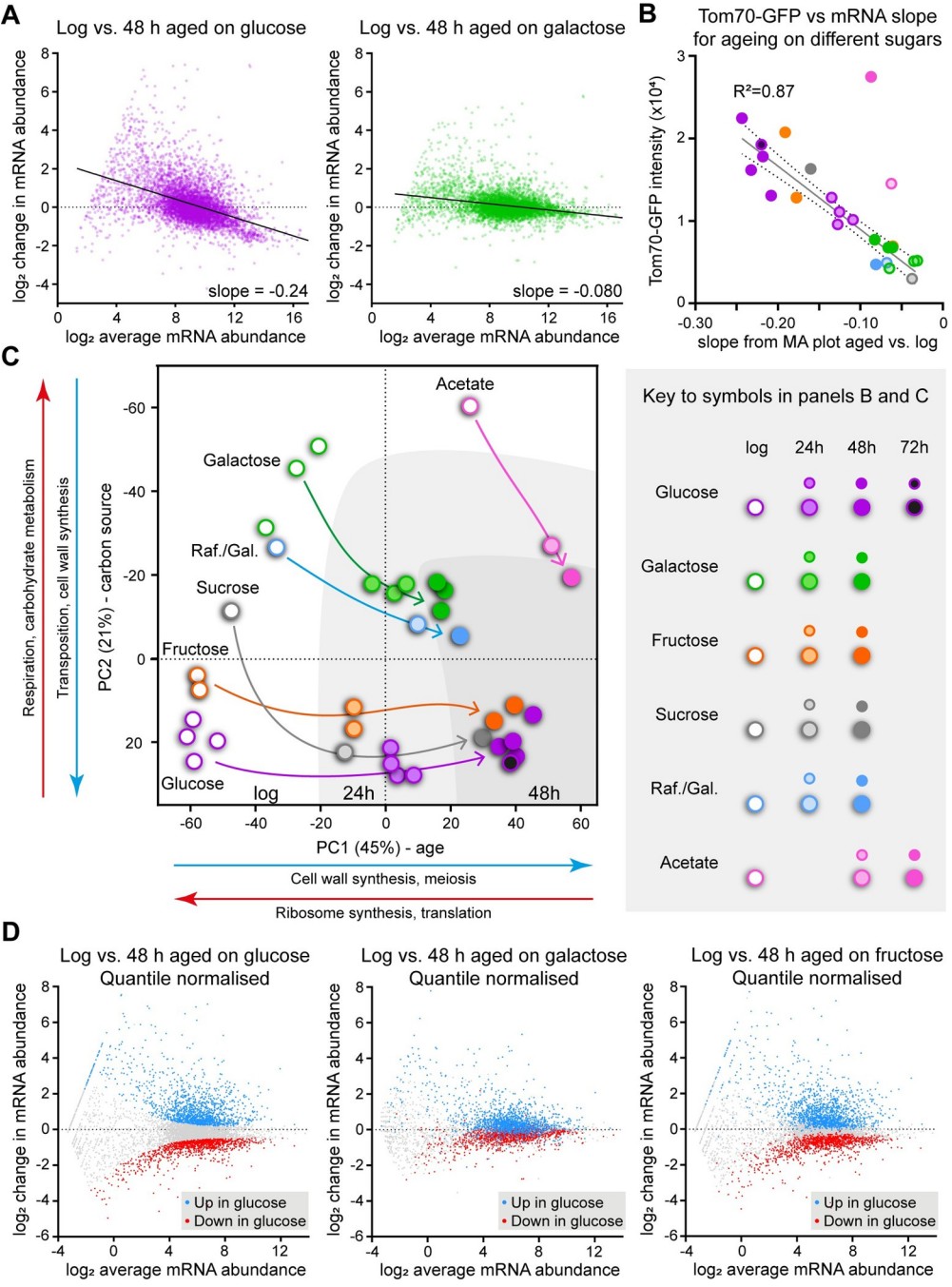

**Fig 2. Effects of diet on gene expression change during ageing.** (A) MA plots showing change in abundance between log and 48 h relative to average abundance for each mRNA, for cells growing on glucose (left) or galactose (right). Positive values on y-axis indicate increased abundance with age relative to average, negative values indicate decreased abundance, and zero indicates no change. Abundances are size-factor normalised log$_2$ transformed read counts from poly(A) selected sequencing libraries, genes with <4 counts at log phase were filtered as changes cannot be accurately quantified for these. Data is average of 2 biological replicates, slopes calculated by linear regression. (B) Plot of Tom70-GFP intensity against slope values of mRNA distribution change for multiple biological replicate samples of wild-type cells aged on different carbon sources. For each individual aged sample, half the cells were used for RNAseq and slope calculated by linear regression from an MA plot of gene expression versus matched log phase sample. Tom70-GFP intensity was obtained from imaging flow cytometry performed on the other half of the sample. Line of best fit calculated by linear regression showing 95% confidence window, acetate samples were excluded as they clearly deviate from the trend. Key to carbon source and age given below graph. (C) PCA plot for individual biological replicates at log, 24 and 48 h of age on different carbon sources, with darkening shades of grey background

corresponding to increasing age. Calculated from size-factor normalised $\log_2$ transformed read counts for all genes excluding mitochondrial genome-encoded mRNAs. Major GO categories for each PC are indicated based on the 300 highest rotation genes underlying the PC in either direction, full GO analysis in S1 File. (D) MA plots showing change in abundance between log and 48 h relative to average abundance for each mRNA, highlighting genes that change significantly during ageing in glucose, for cells growing on glucose (left), galactose (middle), or fructose (right). Abundances are size-factor normalised $\log_2$ transformed read counts from poly(A) selected sequencing libraries, datasets were quantile normalised to match overall distributions, and genes with <4 counts at log phase were filtered out. Data is average of 2 biological replicates, significant differential gene expression calculated using DESeq2 with $p < 0.01$. The data underlying this figure can be found in S2 File, full GO analysis for C in S1 File. PCA, principal component analysis.

further towards the low ribosome synthesis/translation end of PC1 than equivalently aged cells on galactose, consistent with the lower Rpl13a-mCherry signal and the lower fitness of glucose-aged cells compared to galactose (Fig 1C and 1F). As shown in the accompanying manuscript [60], the slope change on MA plots during ageing arises from slowing of growth, so the difference in slope between glucose and galactose-aged cells reflects both the faster growth when young and the slower growth when old of cells on glucose compared to cells on galactose.

It is clear from the PCA that PC1 dominates the datasets, but this is tightly associated with the distortion of the whole gene expression profile that we represent in the slope value. This distillation of the whole transcriptome change to a single value is rather reductive and we asked whether subsets of genes are acting differently to the bulk dataset during ageing in different carbon sources. We used quantile normalisation to remove the overall profile difference represented by the slope from the datasets, extracted genes differentially expressed during ageing in glucose then highlighted these genes on MA plots for glucose, galactose, and fructose ageing (Fig 2D). While the normalisation is effective at removing the baseline distortion, it is still obvious that gene expression changes much less during ageing on galactose compared to glucose or fructose. No subsets of genes change in behaviour, and we simply observe that the gene expression changes that occur during ageing in glucose also occur in galactose, just to a much lesser extent.

These experiments show that the onset of the SEP, as marked by Tom70-GFP foci formation, is tightly linked to gene expression dysregulation on fermentable carbon sources. Given that lifespan is little different between glucose and galactose, this means that gene expression dysregulation is an aspect of ageing that is not intrinsically associated with increasing replicative age or lifespan.

## Respiration protects against the SEP

This RNA-seq dataset should contain mechanistic information about healthy ageing. Although the global differences in mRNA distribution in our primary datasets violate a basic assumption of packages such as DEseq2, the quantile normalised datasets are suitable for differential expression analysis. Pairwise normalised glucose versus galactose comparisons at each age yielded variable gene sets, so we asked the more conservative question of which genes are consistently differentially expressed across all age comparisons (Fig 3A and S1 File). This much smaller set is enriched for respiration, which is at first sight surprising as galactose and glucose are both fermentable carbon sources. However, whereas respiration is repressed in glucose, it does occur on galactose and is in fact required for normal growth of S288-derived MEP strains [61–63].

Blocking the electron transport chain to prevent respiration, for example, by deleting *COX9*, impairs growth in galactose but this defect can be offset by removing the transcriptional repressor Gal80 [64]. Unlike wild-type, *cox9Δ gal80Δ* cells ageing in galactose accumulated

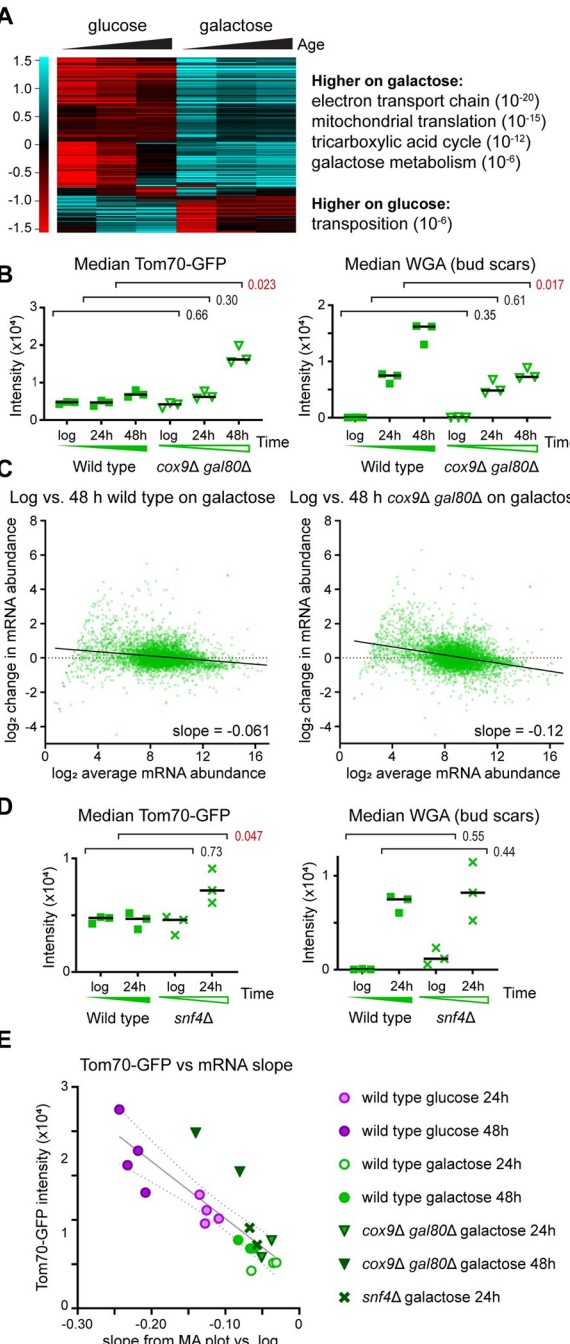

**Fig 3. Respiration is required to avoid the SEP and gene expression dysregulation on galactose.** (A) Hierarchical cluster analysis of genes significantly ($p < 0.01$ by DESeq2 on quantile normalised data) differentially expressed between glucose and galactose at all time points. Average of 2 biological replicates. Principle GO categories with $p$ values are given for each cluster, full GO analysis in S1 File. (B) Imaging flow cytometry for Tom70-GFP and WGA of wild-type and $cox9\Delta$ $gal80\Delta$ populations aged on galactose. Performed and analysed as in Fig 1C, $n = 3$. Rpl13a-mCherry data is given in S3A Fig. (C) MA plots comparing log and 48 h-aged populations of wild type and $cox9\Delta$ $gal80\Delta$ on galactose, performed as in Fig 2A, $n = 1$ for wild type (matched biological replicate to mutant), $n = 2$ $cox9\Delta$ $gal80\Delta$. (D) Imaging flow cytometry for Tom70-GFP and WGA of wild-type and $snf4\Delta$ populations aged on galactose. Performed and analysed as in Fig 1C, $n = 3$. Rpl13a-mCherry data is given in S3A Fig. (E) Plot of Tom70-GFP intensity against slope values of mRNA distribution change, as Fig 2B. The data underlying this figure can be found in S2 File, full GO analysis for A in S1 File. SEP, senescence entry point; WGA, wheat germ agglutinin.

substantial Tom70-GFP at 48 h, despite reaching a much lower replicative age (Figs 3B and S3A), and also underwent greater gene expression dysregulation (Fig 3C). The gene expression phenotype was weaker than wild type after 48 h on glucose but that is probably due to the reduced age of the mutant. Deletion of individual ETC components could drive mitochondrial mis-function rather than simply preventing respiration so we also blocked mitochondrial biogenesis by removing the regulatory Snf4 subunit of the upstream Snf1 complex (yeast AMPK) [65]. Loss of Snf4 did not substantially impair log phase growth in galactose and we were able to purify a 24 h fraction (albeit with very low cell number as cells died very young), and snf4Δ mutants accumulated more Tom70-GFP than wild-type during 24 h replicative ageing in galactose (Figs 3D and S3B). A plot of Tom70-GFP against mRNA slope shows that cox9Δ gal80Δ on galactose approaches wild type on glucose at 48 h despite little effect at 24 h, while snf4Δ at 24 h shows an increase in both parameters compared to wild type at 24 h on galactose, though far less than wild type on glucose at 24 h (Fig 3E).

Taken together, these results show that respiration protects cells against the SEP and against gene expression dysregulation, as well as being required for normal lifespan.

## Respiration can protect a subset of cells during ageing in glucose

Yeast can be forced to respire on glucose by overexpression of the mitochondrial biogenesis transcription factor Hap4, which is known to favour ageing on a longer-lived trajectory with rapid cell division to very old age [35,46,66]. In keeping with this report, median Tom70-GFP was significantly lower at 24 h in MEP cells on glucose overexpressing 1 chromosomal copy of *HAP4* from a *GPD* promoter (Fig 4A), but the difference was smaller and nonsignificant by 48 h while dysregulation of gene expression was not noticeably rescued (S4A Fig). However, the Tom70-GFP profile of 48 h-aged Hap4 overexpressing cells was clearly split into 2 populations, which resolved into a low Tom70-GFP, high WGA population and a high Tom70-GFP, low WGA population akin to the populations observed in wild-type cells by Li and colleagues (Fig 4B) [35]. An equivalent plot of wild-type cells shows that the high WGA, low Tom70-GFP population is largely absent (S4B Fig).

In fact, flow cytometry using a BD Influx cell sorter system detected 3 well-defined fractions in 48 h-aged Hap4 overexpressing populations: high, medium, and low WGA, which we purified and analysed by RNA-seq (S4C Fig). Compared to log phase, the high WGA fraction maintained a youthful gene expression profile with little dysregulation of gene expression, the medium WGA fraction showed an mRNA abundance profile akin to 48 h-aged wild type, but the low WGA fraction displayed the strongest dysregulation of mRNA abundance we have yet observed (Fig 4C–4E). Even after quantile normalisation, the low and high WGA profiles are too different for meaningful genome-wide comparisons to identify individual differentially expressed genes, so we asked whether Hap4 overexpression or that of downstream factors is maintained with age. *HAP4* mRNA itself decreased with age equivalently between the high and low WGA fractions (S4D Fig), but the abundance of many mRNAs that are induced >4-fold by Hap4 overexpression returns to wild-type levels in the low WGA fraction, showing that Hap4 transcription factor activity is not maintained in these cells and therefore the protective effects of respiration are lost (Fig 4F).

We performed an equivalent experiment in wild-type cells to determine whether this heterogeneity is simply a product of Hap4 overexpression (S4E Fig). Although we could not identify defined populations, mRNA abundance profiles of low and high WGA fractions were closely akin to those of low and high WGA fractions from Hap4 overexpressing cells (S4F Fig). This means that within a population aged for a specific time, the youngest cells—those which have ceased to divide rapidly and can be considered senescent—show aberrant mRNA

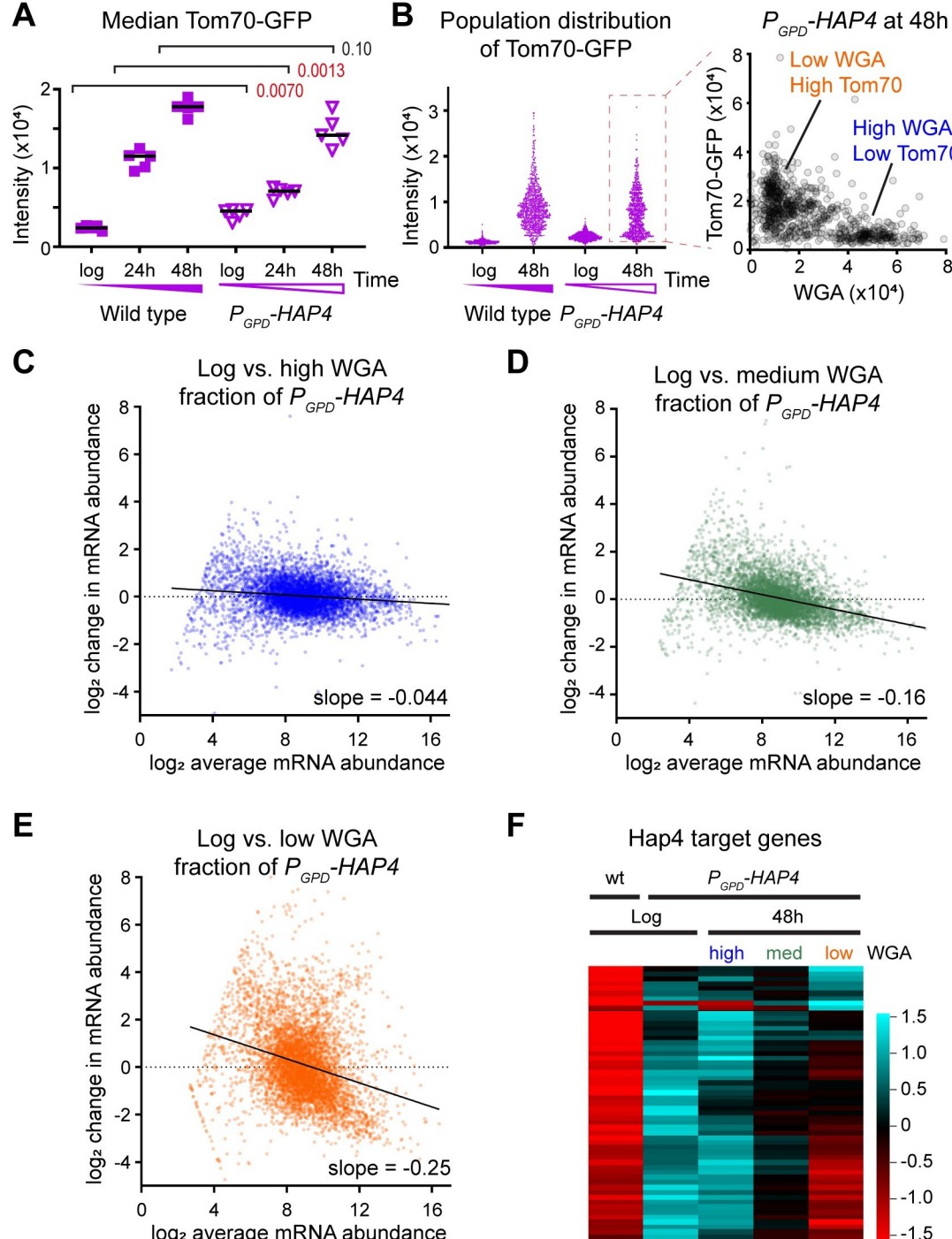

**Fig 4. RNA-seq analysis of cells ageing on different trajectories.** (A) Median Tom70-GFP of wild-type and Hap4-overexpressing cells ($P_{GPD}$-HAP4) on glucose, analysed as in Fig 1C, $n$ = 5. (B) Tom70-GFP population distributions from an individual biological replicate, and plot of Tom70-GFP vs. WGA for the $P_{GPD}$-HAP4 48 h-aged population, highlighting subpopulations demarked by high Tom70-GFP, low WGA (orange) and low Tom70-GFP, high WGA (blue). (C–E) MA plots showing change in mRNA abundance from log phase to the high WGA (C, blue), low WGA (E, orange), or intermediate WGA (D, green) subpopulations of 48 h-aged $P_{GPD}$-HAP4 cells purified by flow cytometry (S4C Fig). Cut-off for low expressed genes on which quantification is unreliable at 16 normalised counts, average of 2 biological replicates. (F) Hierarchical clustering analysis for change in expression of Hap4-regulated genes (defined as those expressed >3-fold higher in $P_{GPD}$-HAP4 compared to wild type at log phase in glucose), for high and low WGA subpopulations of 48 h-aged $P_{GPD}$-HAP4 cells compared to log phase wild-type and $P_{GPD}$-HAP4 controls. Average of 2 biological replicates. The data underlying this figure can be found in S2 File. WGA, wheat germ agglutinin.

abundance patterns, and therefore, gene expression dysregulation must be associated with the SEP rather than ageing itself.

Together, these experiments show that the Hap4 overexpression can increase the fraction of cells that do not undergo SEP onset on glucose, and these cells are marked by youthful gene expression profiles. Combined with the preceding data, this means that respiration during ageing suppresses the SEP and age-linked dysregulation of gene expression.

## Accumulation of a chromosomal fragment is tightly linked to the SEP

Transcriptome data includes expression information for rDNA-encoded ncRNAs (see schematic in S5A Fig), which have a major role in rDNA stability and reflect the activity of silencing factors such as Sir2 [37,67–70]. Gene expression patterns in this region are complex so we defined a set of 4 probes covering each strand of each rDNA intergenic spacer region for quantitation (Fig 5A). These measure overall rDNA ncRNA expression, though the set of ncRNA species detected was very similar between aged samples (S5A Fig). During ageing in glucose, rDNA ncRNA expression rose to a maximum at 24 h before declining slightly at 48 h; while on galactose, these species were lower at 24 h but reached the same level as glucose-aged cells by 48 h (Fig 5A). The ncRNA IGS-1F becomes the most abundant polyadenylated RNA in the cells by 48 h in glucose and galactose (Fig S5B). This high expression reflects a similar accumulation of ERCs (Figs 5B and S5C), which are the primary source of rDNA ncRNA transcripts in aged cells [71], and ERC accumulation follows similar dynamics to rDNA-encoded ncRNAs as expected.

It has been proposed that ERC accumulation is the cause of the nucleolar enlargement that accompanies ageing beginning at the SEP [13,72] so we measured nucleolar size using the RNA polymerase I reporter Rpa190-GFP. In keeping with other markers of the SEP, nucleolar size increased with age on glucose but not on galactose, reinforcing our conclusion that galactose-aged cells do not become senescent (S5D Fig, top). This observation is surprising since ERC levels in 48 h-aged cells are not different between glucose and galactose; combined with our observation that nucleolar enlargement also occurs in cells lacking ERCs [60], we conclude that ERC accumulation cannot be the cause of nucleolar enlargement. Rpa190-GFP signal intensity also did not correlate to nucleolar area and recapitulated behaviour of the ribosomal protein Rpl13a-mCherry marker (compare S5D Fig, bottom panel to Fig 1B, middle panel). On glucose, Rpa190 intensity increases slightly at 24 h but then decreases to below log phase levels in most cells, presumably because cells divide very slowly after the SEP and have a lower demand for ribosomes, whereas on galactose Rpa190 levels change little with age, reflecting a steady requirement for ribosomes.

In the accompanying manuscript, we describe the close association between accumulation of the ChrXIIr fragment (a region of chromosome XII from the rDNA to the centromere-distal telomere) and entry to senescence marked by high Tom70-GFP fluorescence [26,60]. This association emerged from analysis of gene expression, where ChrXIIr accumulation can be readily detected by an average increase in mRNA abundance for genes across this region with age. Expression of genes on ChrXIIr increases on average with age in glucose, and (more slowly) fructose and sucrose, while changes in expression of genes in other genomic regions are unbiased (Fig 5C). In contrast, ChrXIIr accumulation was minimal for cells aged on galactose or a raffinose/galactose mixture, while acetate again showed an intermediate phenotype (Fig 5C). These observations reinforce the connection between ChrXIIr accumulation and the SEP and also mark an association between ChrXIIr and gene expression dysregulation. Furthermore, *cox9Δ gal80Δ* mutants ageing on galactose accumulate almost the same level of

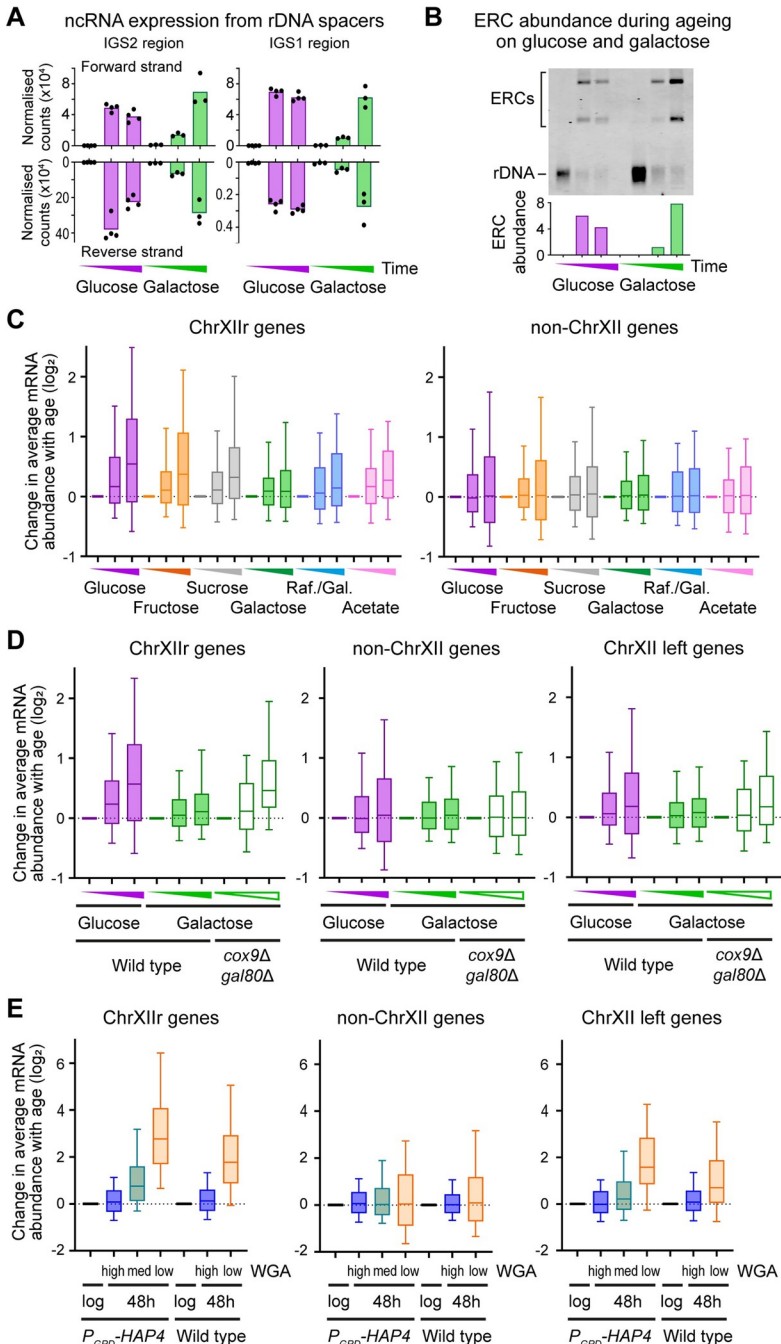

**Fig 5. Accumulation of the ChrXIIr fragment is tightly associated with the SEP.** (A) Quantification of ncRNA transcribed from each strand of the rDNA intergenic spacer regions during ageing on glucose and galactose. Normalised read counts are given for forward and reverse strand transcripts of intergenic spacers IGS1 and IGS2, see S5A Fig for schematic diagram. $n = 4$ for glucose, 3 for galactose. (B) Southern blot analysis of ERCs and chromosomal rDNA in wild-type cells at log, 24 and 48 h on glucose and galactose. Quantification is of the ratio between ERCs and chromosomal rDNA. (C) Log$_2$ change in expression with age of all genes on ChrXIIr [XII:498053–1078177] (left) and all genes except those on chromosome XII and the mitochondrial DNA (right). Boxes show interquartile range with median, whiskers show 10–90th percentiles. $n = 2$ for glucose, fructose, and galactose; $n = 1$ for sucrose, raffinose/galactose mix, and acetate. (D) Change in expression with age of genes on ChrXIIr and controls for wild type on glucose and galactose, and $cox9\Delta$ $gal80\Delta$ on galactose. $n = 2$, analysis as in C. (E) Change in expression for genes on ChrXIIr between purified fractions of $P_{GPD}$-HAP4 and wild-type cells aged 48 h on glucose (Fig 4) relative to log phase controls. $n = 2$ for $P_{GPD}$-HAP4, $n = 1$ for wild type, analysis as in C but note change in scale on y-axis as differences are much greater in these samples. The data underlying this figure can be found in S2 File and S1_Raw_Images.pdf. ERC,

extrachromosomal rDNA circle; ncRNA, non-coding RNA; rDNA, ribosomal DNA; SEP, senescence entry point; WGA, wheat germ agglutinin.

ChrXIIr by 48 h on galactose as wild-type cells ageing on glucose, in keeping with their increased Tom70-GFP and transcriptional dysregulation (Fig 5D).

Importantly, the low and high WGA fractions of 48 h-aged cells overexpressing Hap4 were differentiated by a massive increase and no change in expression of ChrXIIr genes respectively, meaning that even within a population, the presence of ChrXIIr is heterogeneous and specific to the subpopulation of cells that have passed the SEP (Fig 5E). In contrast, IGS transcript levels indicative of ERC abundance were not substantially different between WGA fractions, and non-chromosome XII regions were largely unaffected (Figs 5E and S5E). Furthermore, the same division in expression change of ChrXIIr genes was observed between high and low WGA wild-type cell fractions (Fig 5E), meaning that this effect is not caused by overexpression of Hap4 or downstream targets. We note that expression of genes from the start of Chromosome XII up to the rDNA (ChrXII left) increases marginally with age in glucose though the change is unconvincing (Fig 5D, right), but genes in this region are more highly expressed in the low WGA fraction though still much less than ChrXIIr genes (Fig 5E, right). We interpret this to mean that the same incomplete replication events at mitosis predicted to underlie ChrXIIr formation could also result in complete retention of both Chromosome XII chromatids in the mother cell [60].

These results show that unlike ERCs, accumulation of ChrXIIr is tightly associated with the SEP across a wide range of conditions and mutants, and that dysregulation of gene expression accompanies ChrXIIr accumulation and the SEP.

## Early life events define ageing trajectory on glucose and galactose

Li and colleagues demonstrated that the decision between ageing trajectory for cells in glucose is made early in life [35], and we asked if this principal applies to the time at which diet affects ageing trajectory. We therefore performed media shift experiments, with cells aged for 24 h in glucose or galactose (at which point viability is still >80%, Fig 1D), then harvested and resuspended either in media containing the same or the reciprocal sugar (Fig 6A).

The SEP, as marked by Tom70 intensity, was dependent on early life diet (Figs 6B and S6A): If the first 24 h were spent on glucose, the Tom70-GFP signal observed at 48 h was very high and ageing on galactose for the second 24 h had a significant but not substantial effect; in contrast, if the first 24 h were spent on galactose, ageing on glucose for the second 24 h did not cause an increase in Tom70-GFP signal. Unsurprisingly given the previous results, gene expression dysregulation was strong in populations that spent the first 24 h on glucose and much lower for those that spent the first 24 h in galactose, irrespective of diet for the second 24 h (Fig 6C). Genes significantly differentially expressed during ageing in glucose behaved similarly following transfer to galactose at 24 h, while these genes move in the same direction but to a much lesser extent if the first 24 h of ageing is performed in galactose (S6B Fig). Furthermore, ChrXIIr accumulation was detected only if the first 24 h were spent in glucose (Fig 6D), whereas rDNA ncRNA expression was similar between all 48 h-aged samples, though lower at 24 h in galactose than in glucose (S6C Fig).

It has been reported that ageing cells lose the capacity to properly induce genes, which could explain the importance of the first 24 h [73]. However, the set of genes that are expressed >4-fold higher in galactose than glucose at log, and are presumably critical for metabolic state, are mostly induced to the same level at 48 h irrespective of whether the first 24 h of ageing were spent on glucose or galactose (Fig 6E, left). We see no difference in steady-state levels of

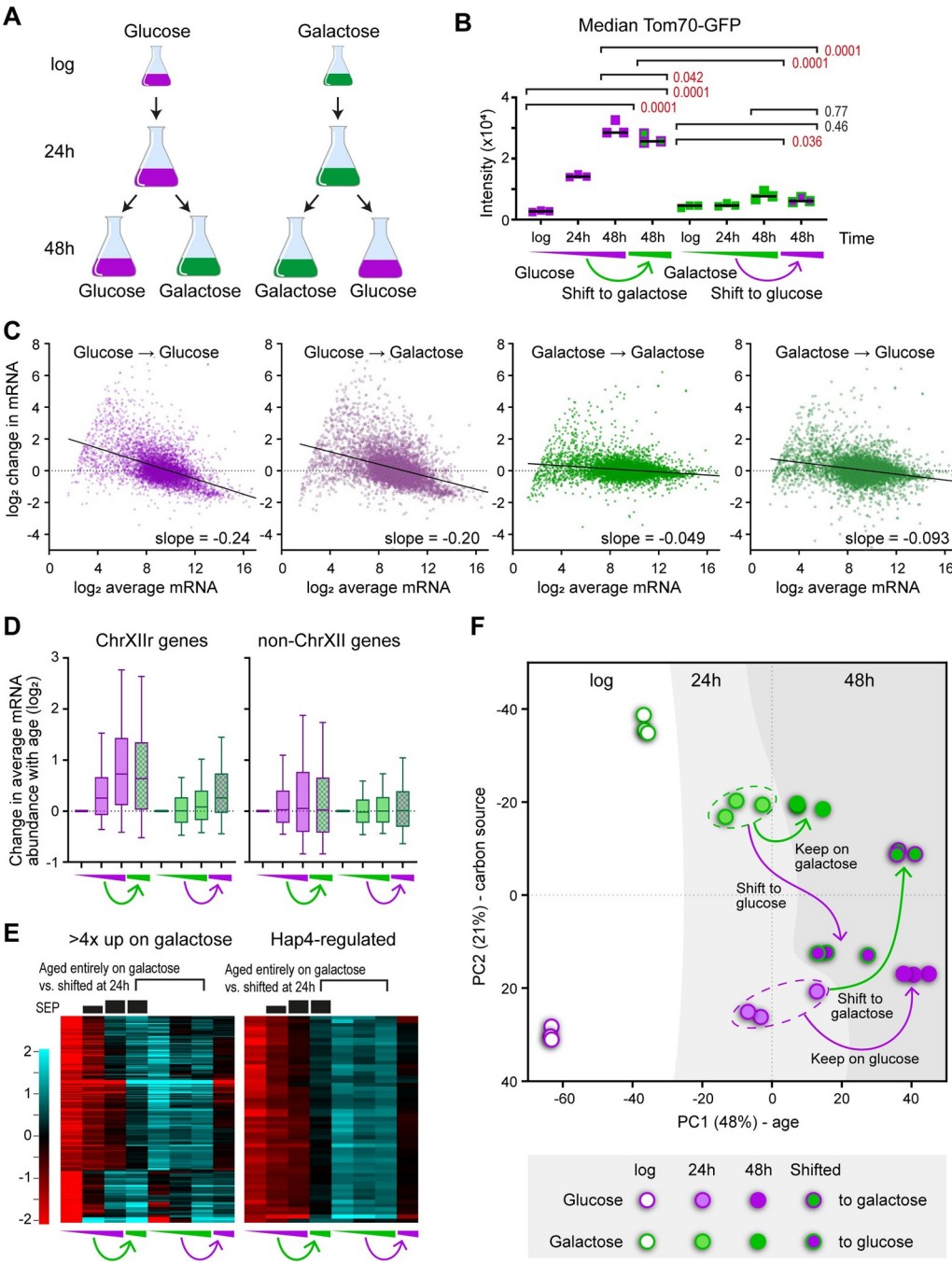

**Fig 6. Media shifts determine the plasticity of ageing phenotypes.** (A) Schematic of experimental set up. Cells are harvested at 24 h then transferred to either the same or the reciprocal media. (B) Median Tom70-GFP intensities for all populations, analysed as in Fig 1C; *p* values from one-way ANOVA with post hoc Tukey test, *n* = 3. (C) MA plots for all four 48 h-aged fractions compared to the relevant log phase controls, performed as in Fig 2A, average of 3 biological replicates. (D) Change in expression with age for genes on ChrXIIr and control set as Fig 5C, average of 3 biological replicates. (E) Hierarchical clustering across all samples of genes either induced >4x on galactose relative to glucose at log phase (left) or of the Hap4-regulated gene set from Fig 4D (right). The extent of SEP entry is indicated diagrammatically above the plots based on Tom70-GFP intensities from B. Average of 3 biological replicates. (F) PCA plot showing changes in mRNA expression pattern, as Fig 2C. The data underlying this figure can be found in S2 File. PCA, principal component analysis; SEP, senescence entry point.

the core galactose regulon of *GAL1*, *GAL10*, *GAL7*, *GAL2* after media shift at 24 h (S6D Fig), though induction may of course be slower, but Hap4 target gene expression was lower in cells shifted from glucose to galactose at 24 h compared to cells that were aged entirely on galactose (Fig 6E, right). Nonetheless, Hap4 target gene expression is not the defining factor for senescence as expression of these genes is if anything slightly higher in glucose-to-galactose shifted cells that exhibit high Tom70-GFP, compared to galactose-to-glucose shifted cells that exhibit low Tom70-GFP (Fig 6E, right).

Globally, a PCA of the transcriptome data divides the influence of diet in the first and second 24 h (Fig 6F). Distance travelled on PC1 is defined by the first 24 h with cells that start on galactose travelling a shorter distance than those which start on glucose irrespective of diet for the second 24 h, so the glucose→glucose and glucose→galactose samples are coincident on PC1, as are the galactose→galactose and galactose→glucose samples. However, PC2 is divided based on diet at point of harvesting, so the glucose log and 24 h are coincident on PC2 with the glucose→glucose and galactose→glucose samples, i.e., it is the diet for the second 24 h that affects PC2. This separation also occurs for samples on galactose at point of harvest; these are all clearly segregated from glucose on PC2 although spread over a larger range of PC2 values. Therefore, highly aged cells largely maintain responsiveness of gene expression to dietary change late in life, but gene expression dysregulation is defined by diet early in life.

Overall, our study shows that diet in early life controls the appearance of age-linked phenotypic changes, even without dietary restriction. This arises through respiration suppressing onset of the SEP, with successful induction of respiration redirecting cells into a pathway that minimises key hallmarks of ageing even on glucose.

## Discussion

The ability to redirect cells from an unhealthy to a healthy ageing trajectory is the primary aim of research into healthy ageing. Here, we demonstrate that substitution of galactose for glucose suppresses cell-intrinsic processes that lead to a spectrum of seemingly pathological changes during yeast replicative ageing. This does not extend lifespan, showing that the mechanisms which limit lifespan are separable from those that cause senescence.

### Diet-induced transitions between ageing trajectories

It is often assumed that ageing pathology and eventual death are the result of a progressive accumulation of unavoidable damage. However, in yeast we find that senescence is not obligatory and that the extent of replicative lifespan is not dependent on damage accumulation, at least not any form of damage that we have assayed. Instead, our findings are consistent with models in which ageing can follow different trajectories, only some of which are associated with pathology. Other trajectories do not result in immortality, but the cause of death is not the accumulation of senescence marks and remains unknown. Importantly, such trajectories end with loss of viability without an extended period of senescence and can therefore be considered as healthy ageing.

Ageing trajectory is defined early in life for yeast and is enforced by a mechanism resilient to changes in gene expression and metabolic state, since media shifts at midlife do not curtail the onset of the SEP, gene expression dysregulation, or ChrXIIr accumulation. Extrachromosomal DNA retained in the mother cell such as ERCs or ChrXIIr offer an insufficient explanation: ERCs do not correlate to any of these phenotypes, and ChrXIIr continues to accumulate after the media shift so is likely a consequence rather than a cause of the trajectory. We doubt that any epigenetic mark at the rDNA or elsewhere would be sufficiently stable or asymmetrically segregated to maintain this trajectory. Loss of vacuolar acidity has been proposed to drive

ageing in yeast and could force cells permanently into the pathological trajectory, as the vacuole regulates iron availability and a lack of iron causes numerous cellular defects [15,17,41,74]. Iron availability is low in the short-lived trajectory, iron importer subunits are up-regulated both in single cell RNA-seq of slow dividing ageing cells and in our RNA-seq analysis of low WGA cells, and we observe that overexpression of the iron-dependent Hap4 transcription factor cannot maintain expression of target genes in cells on the pathological ageing trajectory [35,36]. These observations show that iron deficiency is a feature of the short-lived, high SEP trajectory, but the reason why respiration is so effective in removing cells from this trajectory, and the reason why only a fraction of wild-type cells on glucose follow this trajectory, remains unclear.

Longer lifespan and improved health span are broadly associated with reduced biosynthetic activity (i.e., slower growth), including mutants affecting TOR signalling, PKA signalling, ribosome synthesis, and growth under dietary restriction [44,45,75]. Although mother cells divide at similar rates on glucose and galactose, population growth is slower in galactose implying that the daughter cells begin life with less biomass resulting in an extended growth period before the first cell cycle. For the mother cell, this would mean less biosynthesis per generation, and the observed improvement in health span is therefore consistent with the idea that reduced biosynthetic activity is beneficial for ageing fitness. Our media shift experiment shows that early life defines ageing trajectory, and we speculate that it may be this difference in growth of newborn daughters prior to the first cell division that defines the ageing trajectory followed for the rest of life.

Unfortunately, even in yeast the relationship between diet and ageing is metastable; results that are highly reproducible within an experimental system are not necessarily reproduced in others. For example, 2%→0.5% caloric restriction robustly extends lifespan in microdissection experiments but does not extend lifespan in microfluidic studies [50,76,77], and similarly, galactose improves ageing cell fitness when assayed by microdissection but not by microfluidics (Aspert and Charvin, personal communication) [51]. From this viewpoint, it is very helpful that the data for ageing trajectories encompasses microfluidics, microdissection and now batch culture experiments performed on different derivatives of a standard genetic background (S288C) by different groups, and is therefore robust across experimental systems. This suggests that different ageing trajectories are an intrinsic part of the yeast ageing process with environmental and genetic manipulations biasing the process towards different trajectories, and therefore, even if the effects of specific environmental changes are not reproducible across all experimental systems, the underlying biology is likely to be consistent.

Although gene expression dysregulation has been widely reported to occur as yeast ages, our findings clearly associate gene expression dysregulation primarily with the SEP. Many of the gene expression changes that occur with age resemble induction of the environmental stress response, an active gene expression remodelling in response to stress or slow growth [32,78], and in retrospect it is not too surprising that the SEP is associated with an ESR-like gene expression signature. We suspect that whatever causes terminal arrest of cells on galactose happens quickly and the cells do not have time to adjust gene expression for the new growth rate, whereas in glucose, the majority of cells have an extended period of slow growth prior to complete arrest, during which the environmental stress response is induced.

In trying to understand ageing trajectories, it is worthwhile considering potential cause and effect relationships between the ageing phenotypes we have examined, both here and in [60]. Formation of intense Tom70-GFP foci in G1 cells post SEP is very tightly associated with ChrXIIr across all our datasets, but given that ChrXIIr will persist throughout the cell cycle, it seems most likely that ChrXIIr formation is a proximal cause of the Tom70-GFP phenotype not vice versa. This idea fits very well with the observation of Neurohr and colleagues that the

transfer of extrachromosomal DNA to daughter cells transmits the senescent phenotype [73]. In contrast, gene expression dysregulation is unaffected by reduced ChrXIIr accumulation in *spt3Δ* and remains low in *rad52Δ* despite abundant ChrXIIr and early SEP onset [60]. This means that gene expression dysregulation lies either upstream or occurs independently of the SEP, but the tight segregation in gene expression dysregulation within populations between SEP and non-SEP cells strongly suggests that gene expression dysregulation is an early mechanistic driver of the SEP. Under such a model, mutations such as *spt3Δ* and *rad52Δ* simply modulate the rapidity with which underlying ageing phenotypes, of which gene expression dysregulation may only be one, manifest in senescence phenotypes.

## The importance of respiration in healthy ageing

The benefits of dietary restriction have been attributed to respiration [46–49], and we find that the health benefits of a galactose diet similarly require an active electron transport chain. Respiration is obligatory in higher eukaryotes, but declining expression of mitochondrial protein genes is the most robust feature of ageing transcriptomes across higher eukaryotes and one that is rapidly reversed under dietary restriction (reviewed in [79]). This is somewhat counterintuitive, as increased respiration should increase levels of damaging reactive oxygen species (ROS), but low levels of ROS may actually be protective through the induction of hormesis (reviewed in [80]).

Although enhanced respiration remains disputed as the mechanism of action for dietary restriction in yeast [81], forced respiration during growth on glucose extends lifespan and promotes more cells to a healthy ageing trajectory [35,46,82]. Our findings reinforce these observations and show that cells pushed into this trajectory by overexpression of Hap4 maintain a youthful gene expression state to very advanced age. Loss of iron-dependent Hap4 activity, which likely results from the loss of iron homeostasis in a fraction of cells during ageing on glucose [35,36,41,83], is observed in Hap4 overexpressing cells that follow the healthy ageing trajectory. However, Hap4 target genes, which include many critical respiration proteins, were not differentially expressed between cells that have shifted carbon source at 24 h despite a dramatically different ageing trajectory. This means that expression of Hap4 target genes is either sufficient but not necessary for cells to enter the long-lived, healthy ageing trajectory, or that Hap4 target gene expression is only necessary early in life.

Across this study and the accompanying manuscript, the molecular change most strongly associated with the SEP is ChrXIIr fragment accumulation. In [60], we propose that ChrXIIr forms as a result of replication fork stalling in the rDNA, which would be promoted in the absence of Sir2. The activity of the histone deacetylase Sir2 is promoted by respiration, though the underlying mechanism of this promotion remains unresolved [46,47,49,84], so it is reasonable to relate rDNA recombination events that form ChrXIIr to a lack of respiration. It should be noted that others have attributed the effect of caloric restriction on lifespan to reduced activity of mTOR rather than respiration [81], but this could have a similar outcome as mTOR signalling promotes rDNA recombination through Sir2 and the homologous histone deacetylases Hst3 and Hst4 [85–87]. Therefore, multiple pathways affected by diet coincide on the rDNA locus, all of which could alter the production of the ChrXIIr fragment that is so tightly linked to the SEP.

Overall, our study shows that a transition to a constitutively healthy ageing trajectory is possible, at least in yeast. Of course, substituting galactose as the primary caloric input is neither achievable nor useful in humans, but our findings suggest that dietary change without restriction can offer paths to ageing health benefits currently only observed under dietary restriction.

## Materials and methods

Detailed and updated protocols are available at https://www.babraham.ac.uk/our-research/epigenetics/jon-houseley/protocols.

### Yeast culture and labelling

Yeast strains were constructed by standard methods and are listed in S1 Table, oligonucleotide sequences are given in S2 Table. Plasmid templates were pFA6a-GFP-KanMX4, pYM-N14, pAW8-mCherry, pFA6a-HIS3, and pFA6a-URA3 [88–91]. All cells were grown in YPx media (2% peptone, 1% yeast extract, 2% sugar) at 30°C with shaking at 200 rpm. Media components were purchased from Formedium and media was sterilised by filtration. MEP experiments were performed as described with modifications [24]: Cells were inoculated in 4 ml YPx (where x is the sugar) and grown for 6 to 8 h then diluted in 25 ml YPx and grown for 16 to 18 h to 0.2 to $0.6 \times 10^7$ cells/ml in 50 ml Erlenmeyer flasks, and $0.125 \times 10^7$ cells per aged culture were harvested by centrifugation (15 s at 13,000 g), washed twice with 125 μl PBS, and resuspended in 125 μl of PBS containing approximately 3 mg/ml Biotin-NHS (Pierce 10538723). Cells were incubated for 30 min on a wheel at room temperature, washed once with 125 μl PBS, and resuspended in 125 μl YPx then inoculated in 125 ml YPx at $1 \times 10^4$ cells/ml in a 250 ml Erlenmeyer flask (FisherBrand FB33132) sealed with Parafilm. Approximately 1 μm β-estradiol (from stock of 1 mM Sigma E2758 in ethanol) was added after 2 h (glucose, fructose), 3 h (galactose, sucrose), or 4 h (potassium acetate). An additional $0.125 \times 10^7$ cells were harvested from the log phase culture while biotin labelling reactions were incubating at room temperature. Cells were harvested by centrifugation for 1 min, 4,600 rpm, immediately fixed by resuspension in 70% ethanol and stored at −80°C. To minimise fluorophore bleaching in culture, the window of the incubator was covered with aluminium foil, lights on the laminar flow hood were not used during labelling, and tubes were covered with aluminium foil during biotin incubation. Lifespan assays and competition assays in the MEP background were performed as previously described [24,51].

### Cell purification

Percoll gradients (1 to 2 per sample depending on harvest density) were formed by vortexing 1 ml Percoll (Sigma P1644) with 110 μl 10× PBS in 2 ml tubes and centrifuging 15 min at 15,000 g, 4°C. Ethanol fixed cells were defrosted and washed once with 1 volume of cold PBSE (PBS + 2 mM EDTA) before resuspension in approximately 100 μl cold PBSE per gradient and layering on top of the pre-formed gradients. Gradients were centrifuged for 4 min at 2,000 g, 4°C, then the upper phase and brown layer of cell debris removed and discarded, and 1 ml PBSE was added, mixed by inversion and centrifuged 1 min at 2,000 g, 4°C to pellet the cells, which were then resuspended in 1 ml PBSE per time point (re-uniting samples where split across 2 gradients). Approximately 25 μl Streptavidin microbeads (Miltenyi Biotech 1010007) were added and cells incubated for 5 min on a wheel at room temperature. Meanwhile, 1 LS column per sample (Miltenyi Biotech 1050236) was loaded on a QuadroMACS magnet and equilibrated with cold PBSE in 4°C room. Cells were loaded on columns and allowed to flow through under gravity, washed with 1 ml cold PBSE, and eluted with 1 ml PBSE using plunger. Cells were re-loaded on the same columns after re-equilibration with approximately 500 μl PBSE, washed and re-eluted, and this process repeated for a total of 3 successive purifications. After addition of Triton X-100 to 0.01% to aid pelleting, cells were split into 2 fractions in 1.5 ml tubes, pelleted 30 s at 20,000 g, 4°C, frozen on N2 and stored at −70°C.

For live purification, cells were pelleted, washed twice with synthetic complete 2% glucose media, then resuspended in 2 ml final volume of the same media and incubated for 5 min on a

rotating wheel with 10 μl MyOne streptavidin magnetic beads (Thermo), isolated using a magnet and washed 5 times with 1 ml of the same media. For colony formation assays, cells were streaked on a YPD plate and individual cells moved to specific locations using a Singer MSM400 micromanipulator. Colony size was measured 24 h later on the screen of the micromanipulator imaging with a 4× objective.

## RNA extraction

Cells were resuspended in 50 μl Lysis/Binding Buffer (from mirVANA kit, Life Technologies AM1560), and 50 μl 0.5 μm zirconium beads (Thistle Scientific 11079105Z) added. Cells were lysed with 5 cycles of 30 s 6,500 ms$^{-1}$/30 s on ice in an MP Fastprep bead beater or for 3 min at power 12 in a Bullet Blender (Thermo Fisher) in cold room, then 250 μl Lysis/Binding buffer was added followed by 15 μl miRNA Homogenate Additive and cells were briefly vortexed before incubating for 10 min on ice. Approximately 300 μl acid phenol: chloroform was added, vortexed, and centrifuged 5 min at 13,000 g at room temperature before extraction of the upper phase, and 400 μl room temperature ethanol and 2 μl glycogen (Sigma G1767) were added and mixture incubated for 1 h at −30°C before centrifugation for 15 min at 20,000 g, 4°C. The pellet was washed with cold 70% ethanol and resuspended in 10 μl water, and 1 μl RNA was glyoxylated and analysed on a BPTE mini-gel, and RNA was quantified using a PicoGreen RNA kit (Life Technologies R11490) or Qubit RNA HS Assay Kit.

A total of 150 ng RNA was used to prepare libraries using the NEBNext Ultra II Directional mRNAseq kit with poly(A)+ purification module (NEB E7760, E7490) as described with modifications: Reagent volumes for elution from poly(T) beads, reverse transcription, second strand synthesis, tailing and adaptor ligation were reduced by 50%; libraries were amplified for 13 cycles using 2 μl each primer per 50 μl reaction before 2 rounds of AMPure bead purification at 0.9× and elution in 11 μl 0.1× TE prior to quality control using a Bioanalyzer HS DNA ChIP (Agilent) and quantification using a KAPA Library Quantification Kit (Roche).

## DNA extraction and Southern blot analysis

Cell pellets were resuspended in 50 μl 0.34 U/ml lyticase (Sigma L4025) in 1.2 M sorbitol, 50 mM EDTA, 10 mM DTT and incubated at 37°C for 45 min. After centrifugation at 1,000 g for 5 min, cells were gently resuspended in 80 μl of 0.3% SDS, 50 mM EDTA, 250 μg/ml Proteinase K (Roche 3115801) and incubated at 65°C for 30 min. Approximately 32 μl 5 M KOAc was added after cooling to room temperature, samples were mixed by flicking, and then chilled on ice for 30 min. After 10 min centrifugation at 20,000 g, the supernatant was extracted into a new tube using a cut tip, 125 μl phenol:chloroform (pH 8) was added and samples were mixed on a wheel for 30 min. Samples were centrifuged for 5 min at 20,000 g, the upper phase was extracted using cut tips, and precipitated with 250 μl ethanol. Pellets were washed with 70% ethanol, air-dried and left overnight at 4°C to dissolve in 20 μl TE. After gentle mixing, 10 μl of each sample was digested with 20 U *Xho*I (NEB) for 3 to 6 h in 20 μl 1× CutSmart buffer (NEB), 0.2 μl was quantified using PicoGreen DNA (Life Technologies), and equivalent amounts of DNA separated on 25 cm 1% 1× TBE gels overnight at 120 V. Gels were washed in 0.25 N HCl for 15 min, 0.5 N NaOH for 45 min, and twice in 1.5 M NaCl, 0.5 M Tris (pH 7.5) for 20 min before being transferred to 20 × 20 cm HyBond N+ membrane in 6× SSC. Membranes were probed using a $^{32}$P-labelled random primed probe or a biotin-labelled RNA probe to the NTS1 rDNA intergenic spacer region in 10 ml UltraHyb (Life Technologies) at 42°C and washed with 0.1× SSC 0.1% SDS at 42°C, using standard methods.

## Sequencing and bioinformatics

Libraries were sequenced by the Babraham Institute Sequencing Facility using a NextSeq 500 instrument on 75 bp single end mode. After adapter and quality trimming using Trim Galore (v0.6.6), RNA-seq data was mapped to yeast genome R64-1-1 using HISAT2 v2.1.0 [92] by the Babraham Institute Bioinformatics Facility. Mapped data was imported into SeqMonk v1.47.0 (https://www.bioinformatics.babraham.ac.uk/projects/seqmonk/) and quantified for $\log_2$ total reads mapping to the antisense strand of annotated open reading frames (opposite strand-specific libraries), excluding the mtDNA and the rDNA locus, but with probes included to each strand of the rDNA intergenic spacer regions. Read counts were adjusted by Size Factor normalisation for the full set of quantified probes [93]. Where indicated, a quantile normalisation was additionally applied using the "Match Distribution" function in Seqmonk.

MA plots were generated in GraphPad Prism (v9.2.0) comparing mean and difference for each gene between 2 conditions. Probes with very low numbers of reads post normalisation in the control condition were filtered (cut-offs given in figure legends) as change between datasets cannot be accurately quantified in this case. Slope values were determined from the same filtered datasets using the lm(difference ∼ mean) function in R. PCA plots, hierarchical clustering plots were calculated within SeqMonk, DESeq2 analysis was performed after Quantile normalisation using the Match Distribution function within SeqMonk. GO analysis of individual clusters performed using GOrilla (http://cbl-gorilla.cs.technion.ac.il/) [94]. Quoted *p* values for GO analysis are FDR-corrected according to the Benjamini and Hochberg method (q-values from the GOrilla output), for brevity only the order of magnitude rather than the full q-value is given [95]. Full GO analyses are provided in S1 File.

All RNA-seq data has been deposited at GEO under accession number GSE207503. Aged RNA-seq data for many mutants not included in this manuscript is also deposited in this accession.

## Flow cytometry

Cell pellets were resuspended in 240 μl PBS and 9 μl 10% triton X-100 containing 0.3 μl of 1 mg/ml Streptavidin conjugated with Alexa Fluor 647 (Life Technologies) and 0.6 μl of 1 mg/ml wheat germ agglutinin (WGA) conjugated with CF405S (Biotium). Cells were stained for 10 min at RT on a rotating mixer while covered with aluminium foil, washed once with 300 μl PBS containing 0.01% Triton X-100, resuspended in 30 μl PBS, and immediately subject to flow cytometry analysis. Flow cytometry analysis was conducted using an Amnis ImageStream X Mk II with the following laser power settings: 405 = 25 mW, 488 = 180 mW, 561 = 185 mW, 642 = 5 mW, SSC = 0.3 mW.

Cell populations were gated for single cells based on area and aspect ratio (>0.8) values and in-focus cells were gated based on a gradient RMS value (>50). Further gating of streptavidin positive (AF647) cells was also applied, all in a hierarchical manner and 1,000 events acquired. Before data analysis, compensation was applied according to single-colour controls and a manual compensation matrix creation. Total fluorescence intensity values of different parameters were extracted using the intensity feature of the IDEAS software, with Adaptive Erode modified mask coverage. In the analysis, only positive values of fluorescence were included (i.e., where cells were truly positive for the marker) and median values of populations were determined with Graphpad Prism (v9.2.0). Cell sorting was conducted using a BD Influx system. Cell populations were gated for single cells based on forward and side scatter parameters and aged cells based on streptavidin stain. Distinct populations were sorted based on GFP and WGA-AF405 intensities.

## Statistical analysis

All statistical analysis was performed in GraphPad Prism (v9.2.0).

## Supporting information

**S1 Fig. Supplement to progression of age-linked phenotypes on different diets.** (A) WGA intensities versus manual bud scar counts across ageing on glucose and galactose. (B) Median Vph1-mCherry intensities and areas at different ages on glucose and galactose, as Fig 1C, $n = 3$. (C) Median Tom70-GFP, Rpl13a-mCherry and WGA intensity for populations aged on glucose or fructose, $n = 4$ (log, 24 h), $n = 3$ (48 h), analysed as in Fig 1C. Matching Rpl13a-mCherry and WGA data shown in S1C Fig. (D) Rpl13a-mCherry data for samples in Fig 1G. (E) WGA data for samples in Fig 1G. The data underlying this figure can be found in S2 File. (TIF)

**S2 Fig. Supplement to effects of diet on gene expression change during ageing.** (A) MA plots for gene expression change between 7.5 h and 48 h ageing for wild-type cells on glucose and galactose. Abundances are size-factor normalised $log_2$ transformed read counts from poly (A) selected sequencing libraries, genes with <4 counts at log phase were filtered as changes cannot be accurately quantified for these. Slopes were calculated by linear regression. (B) MA plots for gene expression change between log phase and 48 h ageing for wild-type cells on fructose, sucrose, or a 2% raffinose 0.5% galactose mixture, and equivalent plot for cells aged for 72 h on 2% potassium acetate. Abundances are size-factor normalised $log_2$ transformed read counts from poly(A) selected sequencing libraries, genes with <4 counts at log phase were filtered as changes cannot be accurately quantified for these. Slopes were calculated by linear regression. The data underlying this figure can be found in S2 File. (TIF)

**S3 Fig. Supplement to respiration is required to avoid the SEP and gene expression dysregulation on galactose.** (A) Rpl13a-mCherry data for samples in Fig 3B. (B) Rpl13a-mCherry data for samples in Fig 3D. The data underlying this figure can be found in S2 File. (TIF)

**S4 Fig. Supplement to RNA-seq analysis of cells ageing on different trajectories.** (A) MA plots comparing 48 hour-aged on glucose to log phase control samples for wild type (left) and $P_{GPD}$-*HAP4* (right). Average of 2 biological replicates. (B) Plot of Tom70-GFP versus WGA for wild-type 48 hour-aged population, highlighting subpopulations demarked by high Tom70-GFP, low WGA (orange) and low Tom70-GFP, high WGA (blue). (C) Flow sorting strategy and gating for $P_{GPD}$-*HAP4* cells aged for 48 h on glucose. (D) *HAP4* mRNA abundance in $P_{GPD}$-*HAP4* 48 hour-aged cells compared to wild-type controls, $n = 2$ biological replicates. (E) Flow sorting strategy and gating for wild-type cells aged for 48 h on glucose. (F) MA plots showing change in mRNA abundance from log phase to the high WGA (left, blue) or low WGA (right, orange) subpopulations of 48 hour-aged wild-type cells purified by flow cytometry in E. Cut-off for low expressed genes on which quantification is unreliable at 16 normalised counts; $n = 1$. The data underlying this figure can be found in S2 File. (TIF)

**S5 Fig. Supplement to accumulation of the ChrXIIr fragment is tightly associated with the SEP.** (A) RNA read counts for each strand in 10 bp windows across rDNA intergenic spacer regions, for 2 biological replicate samples each on aged for 48 h on glucose or galactose. Previously described promotors and RNA species are indicated, along with deviations from expected profiles. (B) $Log_2$ normalised read counts for all genes including the rDNA intergenic

spaces. The individual dot in each sample corresponding to the rDNA intergenic spacer probe containing the IGS1-F ncRNA is indicated by a red arrow. (C) Southern blot analysis of ERCs and chromosomal rDNA in 3 biological replicates each of wild-type cells aged for 48 h on glucose and galactose. Quantification is of the ratio between ERCs and chromosomal DNA. (D) Distribution of signal areas and intensities from flow cytometry images of Rpa190-GFP in populations of log phase cells or mother cells aged for 24 and 48 h in YP with 2% glucose or galactose. A total of 1,000 cells are imaged per population after gating for circularity (all) and biotin (aged cells only, based on streptavidin-Alexa647). Images are post-filtered for focus of Rpa190-GFP, leaving approximately 600 cells per population. (E) rDNA ncRNA abundances in the sorted fractions of $P_{GPD}$-HAP4 and wild-type cells from Fig 4. The data underlying this figure can be found in S2 File and S1_Raw_Images.pdf.
(TIF)

**S6 Fig. Supplement to media shifts determine the plasticity of ageing phenotypes.** (A) Rpl13a-mCherry and WGA intensity data for samples in Fig 6B. (B) MA plots showing change in abundance between log and 48 h relative to average abundance for each mRNA, highlighting genes that change significantly during ageing in glucose. Abundances are size-factor normalised $\log_2$ transformed read counts from poly(A) selected sequencing libraries, datasets were quantile normalised to match overall distributions, and genes with <4 counts at log phase were filtered out. Data is average of 3 biological replicates, significant differential gene expression calculated using DESeq2 with $p < 0.01$. (C) rDNA ncRNA abundances for media shift samples. (D) Hierarchical clustering across all samples of genes either induced >16× on galactose relative to glucose at log phase, with primary galactose metabolic genes annotated. The extent of SEP entry is indicated diagrammatically above the plots based on Tom70-GFP intensities from B. Average of 3 biological replicates. The data underlying this figure can be found in S2 File.
(TIF)

**S1 Table. Strains used in this work.** All strains are diploid derivatives of the MEP system [52]. TOM70-GFP, VPH1-mCherry, and RPL13a-mCherry markers are heterozygous to avoid growth defect.
(DOCX)

**S2 Table. Oligonucleotides used for making strains.**
(DOCX)

**S1 File. Full GO analysis for differentially expressed gene sets.**
(XLSX)

**S2 File. Numerical data underlying figures.**
(ZIP)

**S1 Raw images. Full unmodified gel images underlying figure panels.**
(PDF)

## Acknowledgments

We would like to thank Rachael Walker, Attila Bebes, and Aleksandra Lazowska-Addy of the Babraham Institute Flow Facility for their assistance with imaging flow cytometry and sorting of yeast cells; Paula Kokko-Gonzales, Nicole Forrester, and Amelia Edwards of the Babraham Institute Next Generation Sequencing facility for sequencing samples and QC assistance; and Simon Andrews, Anne Segonds-Pichon, and Felix Krueger of the Babraham Institute

Bioinformatics Facility for help with interpreting sequencing data, statistical analysis, and sample processing. We thank Grazia Pizza for southern blot analysis, Gilles Charvin and Théo Aspert for helpful discussions, Andre Zylstra and Della David for critical reading, and Patrick Lindstrom and Dan Gottschling for provision of the MEP system.

## Author Contributions

**Conceptualization:** Dorottya Horkai, Jonathan Houseley.

**Funding acquisition:** Dorottya Horkai, Jonathan Houseley.

**Investigation:** Dorottya Horkai, Hanane Hadj-Moussa, Alex J. Whale.

**Methodology:** Dorottya Horkai, Hanane Hadj-Moussa, Alex J. Whale.

**Visualization:** Dorottya Horkai.

**Writing – original draft:** Jonathan Houseley.

**Writing – review & editing:** Dorottya Horkai, Hanane Hadj-Moussa, Jonathan Houseley.

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
