## [Editor Report · Decision Letter 0]

26 Aug 2022

Dear Jon, 

Thank you for submitting your manuscript entitled "Dietary change without caloric restriction maintains a youthful profile in ageing yeast" for consideration as a Research Article by PLOS Biology.

Your manuscript has now been evaluated by the PLOS Biology editorial staff, as well as by an academic editor with relevant expertise, and I'm writing to let you know that we would like to send your submission out for external peer review.

Once your full submission is complete, your paper will undergo a series of checks in preparation for peer review. After your manuscript has passed the checks it will be sent out for review. To provide the metadata for your submission, please Login to Editorial Manager (https://www.editorialmanager.com/pbiology) within two working days, i.e. by Aug 31 2022 11:59PM.

Kind regards,

Roli

Roland Roberts, PhD

Senior Editor

PLOS Biology

rroberts@plos.org

---

## [Decision Letter · Decision Letter 1]

31 Oct 2022

Dear Jon,

Thank you for your patience while your manuscript "Dietary change without caloric restriction maintains a youthful profile in ageing yeast" was peer-reviewed at PLOS Biology. Your manuscript has been evaluated by the PLOS Biology editors, an Academic Editor with relevant expertise, and by four independent reviewers.

You'll see that the reviews are broadly positive, but between them the reviewers raise a number of substantial concerns. Reviewer #1 only has textual and presentational requests. Reviewer #2 is positive, but wonders about the generalisability to other species and thinks that the causality and mechanisms involved are unclear (s/he has a list of 9 questions, some of which would need experimental data to address them). Reviewer #3's main concern is the reliance on transcriptomic data, which may be masking several possibilities; s/he suggests several experiments to distinguish between them and to address other issues with interpretation. Reviewer #4 shares several of the previous reviewer's concerns, and wants better aging markers.

As you will see in the reviewer reports, which can be found at the end of this email, although the reviewers find the work potentially interesting, they have also raised a substantial number of important concerns. Based on their specific comments and following discussion with the Academic Editor, it is clear that a substantial amount of work would be required to meet the criteria for publication in PLOS Biology. However, given our and the reviewer interest in your study, we would be open to inviting a comprehensive revision of the study that thoroughly addresses all the reviewers' comments. Given the extent of revision that would be needed, we cannot make a decision about publication until we have seen the revised manuscript and your response to the reviewers' comments. Your revised manuscript would need to be seen by the reviewers again, but please note that we would not engage them unless their main concerns have been addressed. 

We appreciate that these requests represent a great deal of extra work, and we are willing to relax our standard revision time to allow you 6 months to revise your study. Please email us (plosbiology@plos.org) if you have any questions or concerns, or envision needing a (short) extension. At this stage, your manuscript remains formally under active consideration at our journal; please notify us by email if you do not intend to submit a revision so that we may withdraw it.

**IMPORTANT - SUBMITTING YOUR REVISION**

*Resubmission Checklist*

*Published Peer Review*

*PLOS Data Policy*

Please note that as a condition of publication PLOS' data policy (http://journals.plos.org/plosbiology/s/data-availability) requires that you make available all data used to draw the conclusions arrived at in your manuscript. If you have not already done so, you must include any data used in your manuscript either in appropriate repositories, within the body of the manuscript, or as supporting information (N.B. this includes any numerical values that were used to generate graphs, histograms etc.). For an example see here: http://www.plosbiology.org/article/info:doi%2F10.1371%2Fjournal.pbio.1001908#s5

*Blot and Gel Data Policy*

Best wishes,

Roli

Roland Roberts, PhD

Senior Editor

PLOS Biology

rroberts@plos.org

REVIEWERS' COMMENTS:

Reviewer #1:

review of PBIOLOGY-D-22-01778R1

Replicative aging in yeast is characterized by a set of changes in cellular physiology, including the appearance of foci of the mitochondrial protein, Tom70, and global changes in transcription. Many of these changes occur rather abruptly, late in the aging process, and these have been collectively termed the Senescence Entry Program (SEP). In this manuscript, the authors show that growth in galactose rather than glucose suppresses the SEP, with senescence-associated changes happening either later, to a lesser extent, or both. Despite this suppression of the cellular signs of aging, replicative lifespan is unaffected by the altered diet, thus the result could be summarized as an increase in "healthspan" without a concomitant increase in lifespan. The authors go on to show that (1) suppression of the SEP depends on the ability to respire; (2) forcing respiration in glucose by overexpression of the transcription factor Hap4 also suppresses the SEP, although it does so only in a sub-population of cells; (3) cells that begin life growing in galactose maintain their youthful profile even after being shifted to glucose, but the converse is not true; and (4) levels of a fragment of chrXII that can be generated by replication problems at the ribosomal DNA, increase concurrently with the SEP. One notable aspect of this work is that, although it has long been known that caloric restriction is beneficial with regard to replicative aging, the galactose diet used here is not calorically restricted compared to glucose. The authors are careful to point out that, given that the change in diet does not increase lifespan, it is possible (though perhaps unlikely) that the changes in cellular physiology that they use to monitor cellular health are not the critical changes that drive aging. This work is interesting, carefully executed, and clearly presented, and thus appropriate for publication in PLoS Biology. Below, I offer comments and suggestions that I believe can improve the manuscript. 

1. Assuming that the physiological changes associated with the SEP are detrimental to cellular health, it seems puzzling that cells grown in glucose don't have a shorter replicative lifespan than those grown in galactose. The authors should comment on this in a more concrete fashion than "... showing that the mechanisms which limit lifespan are separable from those that cause senescence". (In other words, if "healthy aging" doesn't prolong lifespan, and yeast don't really have "quality of life", then should it really be called "healthy aging"?)

2. The authors should note in the legend to Figure 2C that the gradations in gray shading in the background correspond to age. Also in this figure, the gray panel on the right should be labeled "key to symbols", since, as it is, it looks like a stand-alone figure. 

3. Figure 4 and its legend contain several errors (e.g. 2 Ds in the figure, no F, legend refers to things as heat maps when they aren't, etc.). 

4. Mention how the ncRNAs at the rDNA correspond (or not) to c-pro and e-pro. (I believe that IGS1-F and IGS1-R are e-pro, and IGS2-R is c-pro.)

5. Figure 5A is confusing. Is the top row the forward strand and the bottom the reverse? Also, the placement of the map of the intergenic spacer region in the middle of the two rows makes it look like the different columns are probing different regions of the two intergenic regions. 

6. The following comment isn't plausible, since the ratio of intracellular NAD+ to NADH is much too high (60-700) for respiration to alter it enough to significantly alter Sir2 activity: "The activity of the histone deacetylase Sir2 is promoted by respiration through the formation of NAD+ and clearance of nicotinamide (Lin et al., 2004), so it is reasonable to relate rDNA recombination events that form ChrXIIr to a lack of respiration."

7. "without restriction" in last sentence of abstract should be "without caloric restriction". 

Reviewer #2: 

In the manuscript the authors explore how different sugar sources impact aging phenotypes in budding yeast. They utilize the MEP, and metrics of SEP such as Tom70-GFP, Rpl13a-mCherry, RNA seq and bud scars which can be measure on thousands of cells with flow cytometry. They identify galactose as a sugar that prevents Tom70 accumulation with age and cells outcompete cells fed glucose (or several other sugars tested). Initial galactose feeding followed by later glucose has a similar effect. They speculate respiration levels form the foundation of the difference, and further explore this idea using cox9, gal80, snf4 deletions, and overexpression of Hap4 to alter respiration. The data seems very robust with many replicates and statistics. The authors propose two different aging trajectories. There is an advance but the generalizability of these findings to other species is not clear. The correlation between respiration, a change in Chr XII and SEP is interesting and significant but the causality and mechanisms involved are unclear.

Comments for the authors consideration:

1. Its somewhat difficult to compare the RNA seq profiles since the number of divisions are not always comparable at the timepoint collected.

2. Its not clear to me why the Hap4 overexpression creates such heterogeneity in outcome. Is it integrated or on a plasmid that is heterogenous in retention/copy number? Why do some cells not retain expression? Do these cells still have the overexpression transgene construct?

3. Is the heterogeneity in the Hap4 experiment essential to the mixed outcome? Would high Tom70 outcompete low Tom70 that presumably have the same genotype (with the exception of the XIIr)?

4. High Tom70 correlates with more gene dysregulation, but how well accepted is gene dysregulation as a metric of aging? The competition assay seems like a better indicator of health, but this experiment is not performed for any of the mutants.

5. Are there differences in respiration levels between the different mutants? This could be informative for interpretation.

6. This manuscript helps to establish the correlation between the unusual chromosome structure and SEP, but Zylstra et al breaks this correlation with the strain that does not bear the unusual chromosome XIIr but still displays SEP.

7. The authors use "non-ChrXIIr genes" as a control in several experiments, but it would be important to report also on only the rest of ChrXII as a control, since using the entire genome could easily wipe out non-euploid metrics.

8. The authors should comment on why respiration might protect against the SEP. Its counterintuitive since respiration is thought to generate ROS and DNA damage, and DNA damage is assumed to be problematic and a feature of aging.

9. Some mutations do extend lifespan although the manipulations in this manuscript do not. How do those mutants fit with the ideas presented?

Minor

Figure 2, color shades for 48 and 72 hours are not distinguishable

P3, "purified purified"

Reviewer #3:

In the manuscript "Dietary change without caloric restriction maintains a youthful profile in ageing yeast" Horkai and Houseley compare aging trajectories of budding yeast cells grown on different carbon sources. Using purified populations of aged mother cells, they show that cells grown on galactose do not accumulate aggregated mitochondria, a hallmark of aging when cells are grown in glucose. Furthermore, the transcriptome of cells aged in galactose undergoes less changes during aging. Forced expression of the mitochondria biogenesis factor HAP4 in glucose partially induces the aging trajectory typically observed in galactose, suggesting that respirative growth slows cell senescence. Interestingly, they find that the aging trajectories are defined by the nutritional regime cells are grown at young age and cannot be reversed in old mother cells. Finally, the authors find that yeast populations that age more rapidly have increased expression from a chromosome fragment on the right arm of Chromosome XII (XIIr). In an accompanying paper from the same group they show that XIIr is tightly associated with cell senescence.

The manuscript presents a number of interesting observations, such as a suppression of age related phenotypes by a change of diet. While the authors point out that this nutrient regime does not constitute caloric restriction, cell growth in galactose is much (50%) slower than in glucose and thus the observation falls in line with many previous observations showing that slow growth slows replicative aging. Accumulation of a fragment of chromosome XII has been reported before, but the tight association of ChrXIIr accumulation with cell senescence is interesting and suggests a direct link between these two processes. The observation that respiration, and in particular forced expression of HAP4, changes the aging trajectory has been made previously as acknowledged by the authors. While I think these observations are of interest to the readers of PLOS Biology, I have a number of concerns with the manuscript. In particular the interpretation of the transcriptome data seems complicated by factors that are not controlled for as outlined below. The first point applies to both manuscripts and parts of it are a direct copy.

1) My main concern is the interpretation of the transcriptome data. The indicated age dependent decrease in ribosome biogenesis genes is reminiscent of activation of the environmental stress response (ESR), which is a hallmark of aging, slow growth and permanent cell cycle arrest (PMID: 22498653, PMID: 17105650, PMID: 30739799). A terminal cell cycle arrest is entered at the end of replicative life irrespective of the reason of why and how fast cells aged. The fraction of terminally arrested cells in each population of purified aged mother cells could thus dominate the transcriptome profile. This is for example evident in the Hap4 overexpression data in Figure 4: Cells that have undergone only few divisions (low WGA population) have likely been arrested for many hours and display a strong change in gene expression on the MA plots (ESR?) while cells that are still cycling (high WGA population) have not been arrested and show a weak gene expression change. The authors need to control their transcriptome data for this effect by A) determining division times and the fraction terminally arrested cells in the aged populations and B) comparing what part of the gene expression change is driven by activation of an ESR, for example by comparing to published data sets of slow growing or arrested cells. If the gene expression profile in aged populations is a consequence of slow growth or prolonged cell cycle arrests, it changes the interpretation of several figures. For example:

a. Figure 2A, C: PC1 (aging) component correlates with slow growth rate (log phase cells in galactose grow 50% slower than cells grown in glucose, cells grown in acetate grow even slower and already show an "aged (PC1)" trascriptome). As a consequence, cells that already grow slowly when young will show a smaller shift in PC1 as they age (this includes also the acetate data) and a flatter slope in the MA plots in 2A. This is different from the interpretation that growth in galactose prevents age induced gene expression changes.

b. Figure 4C-D: Low WGA cells have undergone fewer divisions than high WGA strains. These low WGA strains have therefore probably been in a terminal cell cycle block for many hours. This data strongly suggests that the slope in the MA plot is a consequence of terminal cell cycle blocks. Because low WGA cells also loose the Hap4 induced gene expression signature, the authors conclude from this data that HAP4 overexpression and respiration are protecting cells from entering senescence. An alternative interpretation is that HAP4 over-expression leads to premature permanent cell cycle exit in about 50% of cells (=low WGA cells). To distinguish between these possibilities, the Tom70-GFP vs WGA plot (4B, right) should be shown for both wild type and PGPD:HAP4 cells. 

2) Cells grown on YP galactose grow roughly 50% slower than on YP glucose. It is therefore surprising that these cells age at the same rate (as suggested by the WGA staining in Figure 1B). The authors should address this issue more carefully, for example with manual bud scar counting at the different timepoints or by performing pedigree analysis.

3) The fact that cells aged on galactose maintain fitness (Tom70-GFP, fitness assay) even though they age at the same rate (viability, WGA staining) as cells aged on glucose is surprising and somewhat contradictory. It would be interesting to repeat the competition assay after 48 hours of aging where the difference between the two conditions (SEP marker Tom70-GFP) seems to be even more exaggerated. Ideally this experiment would be combined with manual bud scar counting, which is more accurate than WGA intensity, especially if cell size differs between conditions.

4) The key definition of the SEP is a lengthening of cell division time. Tom70-GFP is one of several markers (as is nucleolar enlargement). The use of Tom70-GFP as senescence marker is especially problematic in strains that carry mutations that affect mitochondria biogenesis (snf4Δ, PGPDHAP4). As alternative, the authors should measure cell cycle durations of aged mother cells aged in Glc or Gal to determine whether or not they have passed SEP (since increase in cell cycle duration is the key defining feature of the SEP).

5) The WGA staining of snf4Δ mutant cells is high in both log and aged cells. Does that mean that snf4Δ mutant cells are born old or that WGA intensity is not in all cases a good measure for replicative age? This is not discussed in the text.

6) Figure 5B: The ncRNA expression at the 48h timepoint is similar between Glc and Gal and so are the levels of ERCs. The more interesting timepoint in this regard would be the 24h timepoint. From the ncRNA expression I would predict that ERCs accumulate more slowly in Galactose - which would be consistent with the hypothesis that slow growth affects ERC formation and through that aging.

Reviewer #4:

In this manuscript, Horkai and Houseley report that growth on galactose media alleviates the impact of aging on yeast. They further report that the reduced impact is preserved even if the media is changed from galactose to glucose later in life. As their main aging markers, the authors use Tom70-GFP fluorescence as an SEP marker as well as gene expression levels/dysregulation measured by RNAseq.

While the idea of this manuscript is interesting, I have major concerns about the selection of aging markers, interpretation of the data and the conclusions reached. 

1. To convincingly prove the "healthspan" impact of galactose over glucose AND the early galactose exposure's preserving its effect later in life, the authors must use more well-established aging markers and show that the markers are reduced in amount/intensity and/or onset for appearance in cells. 

2. Unless I missed it, the authors don't directly show the budding interval comparisons between galactose and glucose environments in diploid cells. A microfluidic device could readily provide that data. Without this growth rate information, 24hr and 48 hr time points could mean or correspond to different age points in each media or when the two media were swapped.

3. The claimed gene expression changes/dysregulation data is not very convincing or informative. The whole transcriptomic profiles are reduced to slopes of lines. The analyses should ideally be performed in a gene-specific manner across the time points. The bulk RNAseq data plotted/analyzed at the population level is not very informative. The conclusions are drawn from the changes in the slope of the linear fitted line, and this is not very convincing.

4. The authors claim/state in key positions of the manuscript (e.g. abstract) that the yeast lifespan is similar between glucose and galactose. This is not true based on the Liu et al paper cited by the authors. Since galactose lowered yeast lifespan in that paper using a direct single-cell lifespan assay, the concerns raised about this manuscript above (healthspan-related conclusions reached by non-standard aging marker and bulk RNA data analyzed at the slope-of-the-population level) becomes even more important/valid.

5. Taken together, the current state of the manuscript does not support several of the conclusions written. As an example: "Given that lifespan is not different between glucose and galactose, this means that gene expression dysregulation is an aspect of ageing that is not intrinsically associated with increasing replicative age or lifespan." 

This bold statement is simply not true.

---

## [Decision Letter · Decision Letter 2]

23 Jun 2023

Dear Jon,

Thank you for your patience while we considered your revised manuscript "Dietary change without caloric restriction maintains a youthful profile in ageing yeast" for publication as a Research Article at PLOS Biology. This revised version of your manuscript has been evaluated by the PLOS Biology editors, the Academic Editor and the original reviewers.

Based on the reviews, we are likely to accept this manuscript for publication, provided you satisfactorily address the remaining points raised by the reviewers and the following data and other policy-related requests.

IMPORTANT - please attend to the following:

a) Please address the remaining concerns of reviewer #3. The Academic Editor said, "For reviewer 3, I think his/her point is interesting and the authors could address it in the text. As I understand it, even though the galactose mothers are making the same number of buds as the glucose mothers, their daughters are smaller. Because of this, the galactose mothers are building less stuff (growing less) and so their delayed ageing is in line with previous observations of slow growth delaying ageing.

b) In terms of the concerns expressed by reviewer #4, the Academic Editor said, "I agree that not everyone will see 17% shorter lifespan in galactose as being ‘slightly’ shorter and the authors could revise that to say 17%. Still, i think the reviewer puts too much weight on the observations in the Liu paper, especially given that work was done in haploids and this work was done in diploids. While a different approach may show more of a difference between glucose and galactose lifespan than the authors’ approach, I think the authors’ experiments are sufficient to make their claims. The authors’ approaches (bud scars, bulk viability, competitive fitness, colony size) were convincing to me that the galactose cells did not have significantly reduced lifespan/fitness that would affect their conclusions. I also thought that using the measured proxies for the SEP was reasonable and that the authors do not need to measure actual cell division timing.

c) Please address my Data Policy requests below; specifically, we need you to supply the numerical values underlying Figs 1BCDEFG, 2ABCD, 3ABCDE, 4ABCDEF, 5ABCDE, 6BCDEF, S1ABCDE, S2AB, S3AB, S4ABCDEF, S5ABCDE, S6ABCD, either as a supplementary data file or as a permanent DOI’d deposition. I see that you've already provided “Supplementary File S1,” but this only contains data for Figs 2D and 3A (I think?).

d) Please cite the location of the data clearly in all relevant main and supplementary Figure legends, e.g. “The data underlying this Figure can be found in S1 Data” or “The data underlying this Figure can be found in https://doi.org/10.5281/zenodo.XXXXX”

We expect to receive your revised manuscript within two weeks. 

*Published Peer Review History*

*Press*

Sincerely,

Roli

Roland Roberts, PhD

Senior Editor,

rroberts@plos.org,

PLOS Biology

DATA POLICY:

Regardless of the method selected, please ensure that you provide the individual numerical values that underlie the summary data displayed in the following figure panels as they are essential for readers to assess your analysis and to reproduce it: Figs 1BCDEFG, 2ABCD, 3ABCDE, 4ABCDEF, 5ABCDE, 6BCDEF, S1ABCDE, S2AB, S3AB, S4ABCDEF, S5ABCDE, S6ABCD. NOTE: the numerical data provided should include all replicates AND the way in which the plotted mean and errors were derived (it should not present only the mean/average values).

We require the original, uncropped and minimally adjusted images supporting all blot and gel results reported in an article's figures or Supporting Information files. We will require these files before a manuscript can be accepted so please prepare and upload them now. Please carefully read our guidelines for how to prepare and upload this data: https://journals.plos.org/plosbiology/s/figures#loc-blot-and-gel-reporting-requirements

DATA NOT SHOWN?

REVIEWERS' COMMENTS:

Reviewer #1:

[Accept; no comments]

Reviewer #2:

The authors have adequately addressed concerns. 

Reviewer #3:

The authors have shown a good effort and have mostly addressed my concerns. I have raised my concerns about the role of the XIIr in the accompanying manuscript and will not repeat those here. One remaining point the authors should address is the role of different growth rates in driving the gene expression patterns:

My concern about growth rate and aging was partially addressed by showing that mother cells divide at the same rate using bud scar labeling. However, the overall slower population doubling time in poor carbon sources stems from a decreased rate of biomass synthesis, which is completely directed towards daughters during S/G2/M. This results in small daughter cells that take longer in G1 while mother cells apparently can progress through cell division at the same speed as mother cells grown in glucose. Nevertheless, the slow biomass accumulation is characterized by slow a growth transcriptional signature, which reflects reduced ribosome biogenesis and translation rates. Hence my comment that the repression of aging phenotypes in galactose is in line with previous reports about conditions that reduce cell growth delaying aging. Also, my comment about the MA plots was referring to the fact that cells that already start with a slow growth transcriptional signature will have a smaller change in PC1, which is presumably driving the slope in the MA plots. 

Reviewer #4:

I wish I could be more positive, but unfortunately, I am not currently satisfied with some of the responses given by the authors to the major points I had raised. I totally agree with reviewer 3 that "The key definition of the SEP is a lengthening of cell division time". The authors should have measured cell cycle durations of mother cells aged in Glucose or Galactose to determine whether or not they have passed SEP. Liu et al already has the longitudinal single-cell aging data (even though in haploids), the further analysis of which should have direct answers to several of the SEP-related questions posed by the reviewers and currently unaddressed by the authors. When the galactose lowers yeast lifespan by 17% (which is not "slight" as the authors call it), and when the direct/logical reporter of SEP (lengthening of cell division time) is not used during the course of aging, many of the conclusions of this manuscript are still up in the air. 

For example, the first paragraph of the Discussion section reads "Here we demonstrate that substitution of galactose for glucose suppresses cell intrinsic processes that lead to a spectrum of seemingly pathological changes during yeast replicative ageing. This does not extend lifespan, showing that the mechanisms which limit lifespan are separable from those that cause senescence."" This actually limits lifespan and the overall results are not interpreted well in the presence of lifespan reduction in galactose. At the least, this paragraph suggests that there is no lifespan consequence of growth on galactose, when the Liu paper showed that there is. In other words, there are several places in the manuscript that does not explain the counterintuitive results in the presence of 17% lifespan reduction by galactose. On top of that, the SEP is not measured using the metric of the lengthening of cell division time. Taken together, this manuscript needs further work to be complete/definitive.

If youthful profile or healthy aging or healthspan in yeast is defined by the ability of cells to maintain "clock like" low cell-cycle durations generation-after-generation (and this is a very good definition, perhaps the only one), then it is a practical requirement to measure generation-specific cell-cycle durations across media conditions. This manuscript is aiming to address galactose's being better than glucose in facilitating youthful profile. Therefore, anything shorter than measuring cell-cycle durations and SEPs in terms of altered division times would fall short for a strong paper. 

As a final note (that again emerges from the Liu et al paper), it looks like the anti-correlations between cell-cycle durations, their (small) variations across lifespan (as a proxy for cells not entering senescence) and the lifespan itself have been studied at the correlation level by the Acar lab. Going to the single-cell cell-cycle trajectories, grouping "silent" and "noisy" trajectories and looking at their final lifespan could provide further insights into some of these questions.

---

## [Editor Report · Decision Letter 3]

12 Jul 2023

Dear Jon,

Thank you for the submission of your revised Research Article "Dietary change without caloric restriction maintains a youthful profile in ageing yeast" for publication in PLOS Biology. On behalf of my colleagues and the Academic Editor, Sarah Zanders, I'm pleased to say that we can in principle accept your manuscript for publication, provided you address any remaining formatting and reporting issues. These will be detailed in an email you should receive within 2-3 business days from our colleagues in the journal operations team; no action is required from you until then. Please note that we will not be able to formally accept your manuscript and schedule it for publication until you have completed any requested changes.

Sincerely,

Roli

Senior Editor

PLOS Biology

rroberts@plos.org